# Probabilistic Recursive Reasoning for Multi-Agent Reinforcement Learning

**Ying Wen**[§*]**, Yaodong Yang**[§*]**, Rui Luo**[§]**, Jun Wang**[§]**, Wei Pan**[♮]
[§]University College London, [♮]Delft University of Technology
`{ying.wen,yaodong.yang,rui.luo,jun.wang}@cs.ucl.ac.uk`
`{wei.pan}@tudelft.nl`

## ABSTRACT

Humans are capable of attributing latent mental contents such as beliefs, or intentions to others. The social skill is critical in everyday life to reason about the potential consequences of their behaviors so as to plan ahead. It is known that humans use this reasoning ability recursively, i.e. considering what others believe about their own beliefs. In this paper, we start from level-1 recursion and introduce a probabilistic recursive reasoning (PR2) framework for multi-agent reinforcement learning. Our hypothesis is that it is beneficial for each agent to account for how the opponents would react to its future behaviors. Under the PR2 framework, we adopt variational Bayes methods to approximate the opponents' conditional policy, to which each agent finds the best response and then improve their own policy. We develop decentralized-training-decentralized-execution algorithms, PR2-Q and PR2-Actor-Critic, that are proved to converge in the self-play scenario when there is one Nash equilibrium. Our methods are tested on both the matrix game and the differential game, which have a non-trivial equilibrium where common gradient-based methods fail to converge. Our experiments show that it is critical to reason about how the opponents believe about what the agent believes. We expect our work to contribute a new idea of modeling the opponents to the multi-agent reinforcement learning community.

## 1 INTRODUCTION

In the long journey of creating artificial intelligent (AI) that mimics human intelligence, a hallmark of an AI agent is its capabilities of understanding and interacting with other agents (Lake et al., 2017). At the cognitive level, the real-world intelligent entities (e.g. rats, humans) are born to be able to reason about various properties of interests of others (Tolman, 1948; Pfeiffer & Foster, 2013). Those interests usually indicates unobservable mental state including desires, beliefs, and intentions (Premack & Woodruff, 1978; Gopnik & Wellman, 1992). In everyday life, people use this inborn ability to reason about others' behaviors (Gordon, 1986), plan effective interactions (Gallese & Goldman, 1998), or match with the folk psychology (Dennett, 1991). It is known that people can use this reasoning ability recursively; that is, they engage in considering what others believe about their own beliefs. A number of human social behaviors have been profiled by the recursion reasoning ability (Pynadath & Marsella, 2005). Behavioral game theorist and experimental psychologist believe that reasoning recursively is a tool of human cognition that is equipped with evolutionary advantage (Camerer et al., 2004; 2015; Goodie et al., 2012; Robalino & Robson, 2012).

Traditional approach of constructing the models of other agents, also known as opponent modeling, has a rich history in the multi-agent learning (Shoham et al., 2007; Albrecht & Stone, 2018). Even though equipped with modern machine learning methods that could enrich the representation of the opponent's behaviors (He et al., 2016), those algorithms tend to only work either under limited types of scenarios (e.g. mean-field games (Yang et al., 2018)), pre-defined opponent strategies (e.g. Tit-fot-Tat in iterated Prisoner's Dilemma (Foerster et al., 2018)), or in cases where opponents are assumed to constantly return to the same strategy (Da Silva et al., 2006). Recently, a promising methodology from game theory – recursive reasoning – has become popular in opponent modeling

---

*The first two authors have equal contributions. Correspondence to Jun Wang.

(Gmytrasiewicz & Durfee, 2000; Camerer et al., 2004; Gmytrasiewicz & Doshi, 2005; De Weerd et al., 2013b). Similar to the way of thinking from humans, recursive reasoning refers to the belief reasoning process where each agent considers the reasoning process of other agents, based on which it expects to make better decisions. Importantly, it allows an opponent to reason about the modeling agent rather than being a fixed type; the process can therefore be nested in a form as "I believe that you believe that I believe ...". Despite some initial trails (Gmytrasiewicz & Doshi, 2005; Von Der Osten et al., 2017), there has been little work that tries to adopt this idea into the multi-agent deep reinforcement learning (DRL) setting. One main reason is that computing the optimal policy is prohibitively expensive (Doshi & Gmytrasiewicz, 2006; Seuken & Zilberstein, 2008).

In this paper, we introduce a probabilistic recursive reasoning (PR2) framework for multi-agent DRL tasks. Unlike previous work on opponent modeling, each agent here is to consider how the opponents would react to its potential behaviors, before it tries to find the best response for its own decision making. By employing variational Bayes methods to model the uncertainty of opponents' conditional policies, we develop decentralized-training-decentralized-execution algorithms, PR2-Q and PR2-Actor-Critic, and prove the convergence in the self-play scenario when there is only one Nash equilibrium. Our methods are tested on the matrix game and the differential game. The games come with a non-trivial equilibrium where conventional gradient-based methods find challenging. We compare against multiple strong baselines. The results justify the unique value provided by agent's recursive reasoning capability throughout the learning. We expect our work to offer a new angel on incorporating conditional opponent modeling into the multi-agent DRL context.

## 2 RELATED WORK

Game theorists take initiatives in modeling the recursive reasoning procedures (Harsanyi, 1962; 1967). Since then, alternative approaches, including logics-based models (Bolander & Andersen, 2011; Muise et al., 2015) or graphical models (Doshi et al., 2009; Gal & Pfeffer, 2003; 2008), have been adopted. Recently, the idea of Theory of Mind (ToM) (Goldman et al., 2012) from cognitive science becomes popular. An example of ToM is the "Recursive Modeling Method" (RMM) (Gmytrasiewicz et al., 1991; Gmytrasiewicz & Durfee, 1995; 2000), which incorporates the agent's uncertainty about opponent's exact model, payoff, and recursion depth. However, these methods follow the decision-theoretic approaches, and are studied in the limited context of one-shot games. The environment is relatively simple and the opponents are not RL agents.

The Interactive POMDP (I-POMDP) (Gmytrasiewicz & Doshi, 2005) implements the idea of ToM to tackle the multi-agent RL problems. It extends the partially observed MDP (Sondik, 1971) by introducing an extra space of models of other agents into the MDP; as such, an agent can build belief models about how it believes other agents know and believe. Despite the added flexibility, I-POMDP has limitations in its solvability (Seuken & Zilberstein, 2008). Solving I-POMDP with $N$ models in each recursive level with $K$ maximum level equals to solving $\mathbb{O}(N^K)$ PODMPs. Such inherent complexity requires high precision on the approximation solution methods, including particle filtering (Doshi & Gmytrasiewicz, 2009), value iteration (Doshi & Perez, 2008), or policy iteration (Sonu & Doshi, 2015). Out work is different from I-POMDP in that we do not adjust the MDP; instead, we provide a probabilistic framework to implement the recursive reason in the MDP. We approximate the opponent's conditional policy through variational Bayes methods. The induced PR2-Q and PR2-AC algorithms are model-free and can practically be used as the replacement to other multi-agent RL algorithms such as MADDPG (Lowe et al., 2017).

Our work can also be tied into the study of opponent modeling (OM) Albrecht & Stone (2018). OM is all about shaping the anticipated movements of the other agents. Traditional OM can be regarded as level-0 recursive reasoning in that OM methods model how the opponent behaves based on the history, but not how the opponent would behave based on what I would behave. In general, OM methods have two major limitations. One is that OM tends to work with a pre-defined target of opponents; for example, fictitious play (Brown, 1951) and joint-action learners (Claus & Boutilier, 1998) require opponents play stationary strategies, Nash-Q (Hu & Wellman, 2003) require all agents play towards the Nash equilibrium, so do Correlated $Q$-learning (Greenwald et al., 2003), Minimax-Q (Littman, 1994), and Friend-or-foe Q (Littman, 2001). These algorithms become invalid if the opponents change their types of policy. The other major limitation is that OM algorithms require to know the exact (Nash) equilibrium policy of the opponent during training. Typical examples include the series

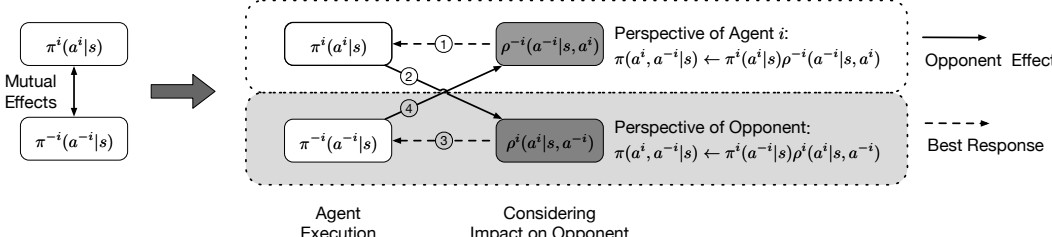

Figure 1: Probabilistic recursive reasoning framework. PR2 decouples the connections between agents by Eq. 3. ①: agent $i$ takes the best response after considering all the potential consequences of opponents' actions given its own action $a^i$. ②: how agent $i$ behaves in the environment serves as the prior for the opponents to learn how their actions would affect $a^i$. ③: similar to ①, opponents take the best response to agent $i$. ④: similar to ②, opponents' actions are the prior knowledge to agent $i$ on estimating how $a^i$ will affect the opponents. Looping from step 1 to 4 forms recursive reasoning.

of *WoLF* models (Bowling, 2005; Bowling & Veloso, 2001a; 2002) or the Nash-Q learning (Hu & Wellman, 2003), both of which require the Nash Equilibrium at each stage game to update the Q-function. By contrast, our proposed methods, PR2-Q & PR2-AC, do not need to pre-define the type of the opponents. Neither do our methods require to know the equilibrium beforehand.

Despite the recent success of applying deep RL algorithms on the single-agent discrete (Mnih et al., 2015) and continuous (Lillicrap et al., 2015) control problems, it is still challenging to transfer these methods into the multi-agent RL context. The reason is because learning independently while ignoring the others in the environment will simply break the theoretical guarantee of convergence (Tuyls & Weiss, 2012). A modern framework is to maintain a centralized critic (i.e. $Q$-network) during training, e.g. MADDPG (Lowe et al., 2017), BiCNet (Peng et al., 2017), and multi-agent soft $Q$-learning (Wei et al., 2018); however, they require strong assumptions that the parameters of agent policies are fully observable, letting alone the centralized $Q$-network potentially prohibits the algorithms from scaling up. By contrast, our approach employs decentralized training with no need to maintain a central critic; neither does it require to know the exact opponents' policies.

## 3 PRELIMINARIES

For an $n$-agent stochastic game (Shapley, 1953), we define a tuple $(\mathcal{S}, \mathcal{A}^1, \ldots, \mathcal{A}^n, r^1, \ldots, r^n, p, \gamma)$, where $\mathcal{S}$ denotes the state space, $p$ is the distribution of the initial state, $\gamma$ is the discount factor for future rewards, $\mathcal{A}^i$ and $r^i = r^i(s, a^i, a^{-i})$ are the action space and the reward function for agent $i \in \{1, \ldots, n\}$ respectively. Agent $i$ chooses its action $a^i \in \mathcal{A}^i$ according to the policy $\pi^i_{\theta^i}(a^i|s)$ parameterized by $\theta^i$ conditioning on some given state $s \in \mathcal{S}$. Let us define the joint policy as the collection of all agents' policies $\pi_\theta$ with $\theta$ representing the joint parameter. It is convenient to interpret the joint policy from the perspective of agent $i$ such that $\pi_\theta = (\pi^i_{\theta^i}(a^i|s), \pi^{-i}_{\theta^{-i}}(a^{-i}|s))$, where $a^{-i} = (a^j)_{j \neq i}$, $\theta^{-i} = (\theta^j)_{j \neq i}$, and $\pi^{-i}_{\theta^{-i}}(a^{-i}|s)$ is a compact representation of the joint policy of all complementary agents of $i$. At each stage of the game, actions are taken simultaneously. Each agent is presumed to pursue the maximal cumulative reward (Sutton et al., 1998), expressed as

$$\max \ \eta^i(\pi_\theta) = \mathbb{E}\left[\sum_{t=1}^{\infty} \gamma^t r^i(s_t, a^i_t, a^{-i}_t)\right], \tag{1}$$

with $(a^i_t, a^{-i}_t)$ sample from $(\pi^i_{\theta^i}, \pi^{-i}_{\theta^{-i}})$. Correspondingly, for the game with (infinite) time horizon, we can define the state-action $Q$-function by $Q^i_{\pi_\theta}(s_t, a^i_t, a^{-i}_t) = \mathbb{E}\left[\sum_{l=0}^{\infty} \gamma^l r^i(s_{t+l}, a^i_{t+l}, a^{-i}_{t+l})\right].$

### 3.1 NON-CORRELATED FACTORIZATION ON THE JOINT POLICY

In the multi-agent learning tasks, each agent can only control its own action; however, the resulting reward value depends on other agents' actions. The $Q$-function of each agent, $Q^i_{\pi_\theta}$, is subject to the joint policy $\pi_\theta$ consisting of all agents' policies. One common approach is to decouple the joint policy assuming conditional independence of actions from different agents (Albrecht & Stone, 2018):

$$\pi_\theta(a^i, a^{-i}|s) = \pi^i_{\theta^i}(a^i|s)\pi^{-i}_{\theta^{-i}}(a^{-i}|s). \tag{2}$$

The study regarding the topic of "centralized training with decentralized execution" in the deep RL domain, including MADDPG (Lowe et al., 2017), COMA (Foerster et al., 2017), MF-AC (Yang et al.,

2018), Multi-Agent Soft-$Q$ (Wei et al., 2018), and LOLA (Foerster et al., 2018), can be classified into this category (see more clarifications in Appendix B). Although the non-correlated factorization of the joint policy simplifies the algorithm, this simplication is vulnerable because it ignores the agents' connections, e.g. impacts of one agent's action on other agents, and the subsequent reactions from other agents. One might argue that during training, the joint $Q$-function should potentially guide each agent to learn to consider and act for the mutual interests of all the agents; nonetheless, a counter-example is that the non-correlated policy could not even solve the simplest two-player zero-sum differential game where two agents act in $x$ and $y$ with the reward functions defined by $(xy, -xy)$. In fact, by following Eq. 2, both agents are reinforced to trace a cyclic trajectory that never converge to the equilibrium (Mescheder et al., 2017).

It is worth clarifying that the idea of non-correlated policy is still markedly different from the independent learning (IL). IL is a naive method that completely ignore other agents' behaviors. The objective of agent $i$ is simplified to $\eta^i(\pi_{\theta^i})$, depending only on $i$'s own policy $\pi_{\theta^i}$ compared to Eq. 1. As Lowe et al. (2017) has pointed out, in IL, the probability of taking a gradient step in the correct direction decreases exponentially with the increasing number of agents, letting alone the major issue of the non-stationary environment due to the independence assumption (Tuyls & Weiss, 2012).

## 4 MULTI-AGENT PROBABILISTIC RECURSIVE REASONING

In the previous section, we have shown the weakness of the learning algorithms that build on the non-correlated factorization on the joint policy. Here we introduce the probabilistic recursive reasoning approach that aims to capture how the opponents believe about what the agent believes. Under such setting, we devise a new multi-agent policy gradient theorem. We start from assuming the true opponent conditional policy $\pi_{\theta^{-i}}^{-i}$ is given, and then move onward to the practical case where it is approximated through variational inference.

### 4.1 PROBABILISTIC RECURSIVE REASONING

The issue on the non-correlated factorization is that it fails to help each agent to consider the consequence of its action on others, which could lead to the ill-posed behaviors in the multi-agent learning tasks. On the contrary, people explicitly attribute contents such as beliefs, desires, and intentions to others in daily life. It is known that human beings are capable of using this ability recursively to make decisions. Inspired by this, here we integrate the concept of recursive reasoning into the joint policy modeling, and propose the new probabilistic recursive reasoning (PR2) framework. Specifically, we employ the nested process of belief reasoning where each agent simulates the reasoning process of other agents, thinking about how its action would affect others, and then make actions based on such predictions. The process can be nested in a form as "I believe [that you believe (that I believe)]". Here we start from considering the level-1 recursion, as psychologist have found that humans tend to reason on average at one or two level of recursion (Camerer et al., 2004), and levels higher than two do not provide significant benefits (De Weerd et al., 2013a;b; de Weerd et al., 2017). Based on this, we re-formulate the joint policy by

$$\pi_\theta(a^i, a^{-i}|s) = \underbrace{\pi_{\theta^i}^i(a^i|s)\pi_{\theta^{-i}}^{-i}(a^{-i}|s, a^i)}_{\text{Agent } i\text{'s perspective}} = \underbrace{\pi_{\theta^{-i}}^{-i}(a^{-i}|s)\pi_{\theta^i}^i(a^i|s, a^{-i})}_{\text{The opponents' perspective}}. \tag{3}$$

Similar ways of decomposition can also be found in dual learning (Xia et al., 2017) on machine translation. From the perspective of agent $i$, the first equality in Eq. 3 indicates that the joint policy can be essentially decomposed into two parts. The conditional part $\pi_{\theta^{-i}}^{-i}(a^{-i}|s, a^i)$ represents what actions would be taken by the opponents given the fact that the opponents know the current state of environment and agent $i$'s action; this is based on what agent $i$ believes other opponents might think about itself. Note that the way of thinking developed by agent $i$ regarding how others would consider of itself is also shaped by opponents' original policy $\pi_{\theta^{-i}}^{-i}(a^{-i}|s)$, as this is also how the opponents actually act in the environment. Taking into account different potential actions that agent $i$ thinks the opponents would take, agent $i$ uses the marginal policy $\pi_{\theta^i}^i(a^i|s)$ to find the best response. To this end, a level-1 recursive procedure is established: $a^i \to a^{-i} \to a^i$. The same inference logic can be applied to the opponents from their perspectives, as shown in the second equality of Eq. 3.

Albeit intuitive, Eq. 3 may not be practical due to the requirement on the full knowledge regarding the actual conditional policy $\pi_{\theta^{-i}}^{-i}(a^{-i}|s, a^i)$. A natural solution is that one approximates the actual

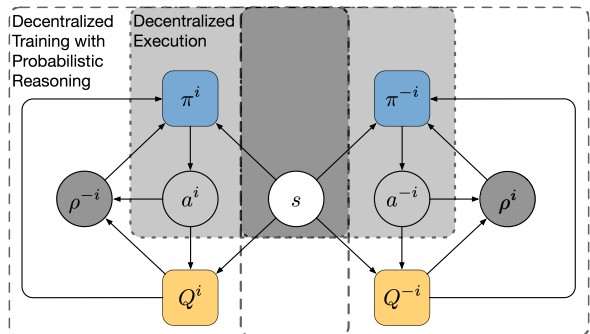

Figure 2: Diagram of multi-agent PR2 learning algorithms. It conducts decentralized training with decentralized execution. The light grey areas on two sides indicate decentralized execution for each agent. White areas give the decentralized learning procedures. All agents share the interaction experiences in the environment represented by dark area in the middle.

policy via a best-fit model from a family of distributions. We denote this family as $\rho_{\phi^{-i}}^{-i}(a^{-i}|s, a^i)$ with learnable parameter $\phi^{-i}$. PR2 is probabilistic as it considers the uncertainty of modeling $\pi_{\theta^{-i}}^{-i}(a^{-i}|s, a^i)$. The reasoning structure is now established as shown in Fig. 1. With the recursive joint policy defined in Eq. 3, the $n$-agent learning task can therefore be formulated as

$$\operatorname*{arg\,max}_{\theta^i, \phi^{-i}} \ \eta^i \left( \pi_{\theta^i}^i(a^i|s) \rho_{\phi^{-i}}^{-i}(a^{-i}|s, a^i) \right), \tag{4}$$

$$\operatorname*{arg\,max}_{\theta^{-i}, \phi^i} \ \eta^{-i} \left( \pi_{\theta^{-i}}^{-i}(a^{-i}|s) \rho_{\phi^i}^i(a^i|s, a^{-i}) \right). \tag{5}$$

With the new learning protocol defined in Eq. 4 and 5, each agent now learns its own policy as well as the approximated conditional policy of other agents given its own actions. In such a way, both the agent and the opponents can keep track of the joint policy by $\pi_{\theta^i}^i(a^i|s)\rho_{\phi^{-i}}^{-i}(a^{-i}|s, a^i) \to \pi_\theta(a^i, a^{-i}|s) \leftarrow \pi_{\theta^{-i}}^{-i}(a^{-i}|s)\rho_{\phi^i}^i(a^i|s, a^{-i})$. Once converged, the resulting approximate satistfies: $\pi_\theta(a^i, a^{-i}|s) = \pi_{\theta^i}^i(a^i|s)\rho_{\phi^{-i}}^{-i}(a^{-i}|s, a^i) = \pi_{\theta^{-i}}^{-i}(a^{-i}|s)\rho_{\phi^i}^i(a^i|s, a^{-i})$, according to Eq. 3.

## 4.2 PROBABILISTIC RECURSIVE REASONING POLICY GRADIENT

Given the true opponent policy $\pi_{\theta^{-i}}^{-i}$ and that each agent tries to maximize its cumulative return in the stochastic game with the objective defined in Eq. 1, we establish the policy gradient theorem by accounting for the PR2 joint policy decomposition in Eq. 3.

**Proposition 1.** *In a stochastic game, under the recursive reasoning framework defined by Eq. 3, the update for the multi-agent recursive reasoning policy gradient method can be derived as follows:*

$$\nabla_{\theta^i} \eta^i = \mathbb{E}_{s \sim p, a^i \sim \pi^i} \left[ \nabla_{\theta^i} \log \pi_{\theta^i}^i(a^i|s) \int_{a^{-i}} \pi_{\theta^{-i}}^{-i}(a^{-i}|s, a^i) Q^i(s, a^i, a^{-i}) \, \mathrm{d}a^{-i} \right]. \tag{6}$$

*Proof. See Appendix B.2.* ∎

Proposition 1 states that each agent should improve its policy toward the direction of the best response after it takes into account all kinds of possibilities of how other agents would react if that action is taken. The term of $\pi_{\theta^{-i}}^{-i}(a^{-i}|s, a^i)$ can be regarded as the **posterior** estimation of agent $i$'s belief about how the opponents would respond to his action $a^i$, given opponents' true policy $\pi_{\theta^{-i}}^{-i}(a^{-i}|s)$ serving as the **prior**. Note that compared to the direction of policy update in the conventional multi-agent policy gradient theorem (Wei et al., 2018), $\int_{a^{-i}} \pi_{\theta^{-i}}^{-i}(a^{-i}|s)Q^i(s, a^i, a^{-i}) \, \mathrm{d}a^{-i}$, the direction of the gradient update in PR2 is guided by the term $\int_{a^{-i}} \pi_{\theta^{-i}}^{-i}(a^{-i}|s, a^i)Q^i(s, a^i, a^{-i}) \, \mathrm{d}a^{-i}$.

In practice, agent $i$ might not have access to the opponents' actual policy parameters $\theta^{-i}$, it is often needed to approximate $\pi_{\theta^{-i}}^{-i}(a^{-i}|s, a^i)$ by $\rho_{\phi^{-i}}^{-i}(a^{-i}|s, a^i)$, thereby we propose Proposition 2.

**Proposition 2.** *In a stochastic game, under the recursive reasoning framework defined by Eq. 3, with the opponent policy approximated by $\rho_{\phi^{-i}}^{-i}(a^{-i}|s, a^i)$, the update for the multi-agent recursive reasoning policy gradient method can be formulated as follows:*

$$\nabla_{\theta^i} \eta^i = \mathbb{E}_{s \sim p, a^i \sim \pi^i} \left[ \nabla_{\theta^i} \log \pi_{\theta^i}^i (a^i|s) \cdot \mathbb{E}_{a^{-i} \sim \rho_{\phi^{-i}}^{-i}} \left[ \frac{\pi_{\theta^{-i}}^{-i}(a^{-i}|s,a^i)}{\rho_{\phi^{-i}}^{-i}(a^{-i}|s,a^i)} Q^i(s,a^i,a^{-i}) \right] \right]. \tag{7}$$

*Proof. Substituting the approximated model $\rho_{\phi^{-i}}^{-i}(a^{-i}|s,a^i)$ for the true policy $\pi_{\theta^{-i}}^{-i}$ in Eq. 6.* ∎

Proposition 2 raises an important point: the difference between decentralized training (algorithms that do not require the opponents' policies) with centralized learning (algorithms that require the opponents' policies) can in fact be quantified by a term of importance weights, similar to the connection between on-policy and off-policy methods. If we find a best-fit approximation such that $\rho_{\phi^{-i}}^{-i}(a^{-i}|s,a^i) \to \pi_{\theta^{-i}}^{-i}(a^{-i}|s,a^i)$, then Eq.7 collapses into Eq. 6.

Based on Proposition 2, we could provide multi-agent PR2 learning algorithm. As illustrated in Fig. 2, it is a decentralized-training-with-decentralized-execution algorithm. In this setting, agents share the experiences in the environment including state and historical joint actions, while each agent receive its rewards privately. Our method does not require the knowledge of other agents' policy parameters. We list the pseudo-code of PR2-AC and PR2-Q in Appendix A. Finally, one last piece missing is how to find the best-fit approximation of $\rho_{\phi^{-i}}^{-i}(a^{-i}|s,a^i)$.

## 4.3 Variational Inference on Opponent Conditional Policy

We adopt an optimization-based approximation to infer the unobservable $\rho_{\phi^{-i}}^{-i}(a^{-i}|s,a^i)$ via variational inference (Jordan et al., 1999). We first define the trajectory $\tau$ up to time $t$ including the experiences of $t$ consecutive time stages, i.e. $\tau = [(s_1,a_1^i,a_1^{-i}),\ldots,(s_t,a_t^i,a_t^{-i})]$. In the probabilistic reinforcement learning (Levine, 2018), the probability of $\tau$ being generated can be derived as

$$p(\tau) = \left[ p(s_1) \prod_{t=1}^{T} p(s_{t+1}|s_t,a_t^i,a_t^{-i}) \right] \exp \left( \sum_{t=1}^{T} r^i(s_t,a_t,a_t^{-i}) \right). \tag{8}$$

Assuming the dynamics is fixed (i.e. the agent can not influence the environment transition probability), our goal is then to find the best approximation of $\pi_{\theta^i}^i(a_t^i|s_t)\rho_{\phi^{-i}}^{-i}(a_t^{-i}|s_t,a_t^i)$ such that the induced trajectory distribution $\hat{p}(\tau)$ can match with the true trajectory probability $p(\tau)$:

$$\hat{p}(\tau) = p(s_1) \prod_{t=1}^{T} p(s_{t+1}|s_t,a_t^i,a_t^{-i}) \pi_{\theta^i}^i(a_t^i|s_t) \rho_{\theta^{-i}}^{-i}(a_t^{-i}|s_t,a_t^i). \tag{9}$$

In other words, we can optimize the opponents' policy $\rho_{\phi^{-i}}^{-i}$ via minimizing the *KL*-divergence, i.e.

$$D_{\mathrm{KL}}(\hat{p}(\tau)\|p(\tau)) = -\mathbb{E}_{\tau \sim \hat{p}(\tau)}[\log p(\tau) - \log \hat{p}(\tau)]$$
$$= -\sum_{t=1}^{t=T} E_{\tau \sim \hat{p}(\tau)} \left[ r^i\left(s_t,a_t^i,a_t^{-i}\right) + \mathcal{H}\left( \pi_{\theta^i}^i\left(a_t^i|s_t\right) \rho_{\phi^{-i}}^{-i}\left(a^{-i}|s_t,a_t^i\right) \right) \right]. \tag{10}$$

Besides the reward term, the objective introduces an additional term of the conditional entropy on the joint policy $\mathcal{H}\left( \pi_{\theta^i}^i\left(a_t^i|s_t\right) \rho_{\phi^{-i}}^{-i}\left(a^{-i}|s_t,a_t^i\right) \right)$ that potentially promotes the explorations for both the agent $i$'s best response and the opponents' conditional policy. Note that the entropy here is conditioning not only on the state $s_t$ but also on agent $i$'s action. Minimizing Eq. 10 gives us:

**Theorem 1.** *The optimal Q-function for agent i that satisfies minimizing Eq. 10 is formulated as:*

$$Q_{\pi_\theta}^i(s,a^i) = \log \int_{a^{-i}} \exp(Q_{\pi_\theta}^i(s,a^i,a^{-i})) \, \mathrm{d}a^{-i}. \tag{11}$$

*And the corresponding optimal opponent conditional policy reads:*

$$\rho_{\phi^{-i}}^{-i}(a^{-i}|s,a^i) = \frac{1}{Z} \exp(Q_{\pi_\theta}^i(s,a^i,a^{-i}) - Q_{\pi_\theta}^i(s,a^i)) \tag{12}$$

*Proof. See Appendix C.* ∎

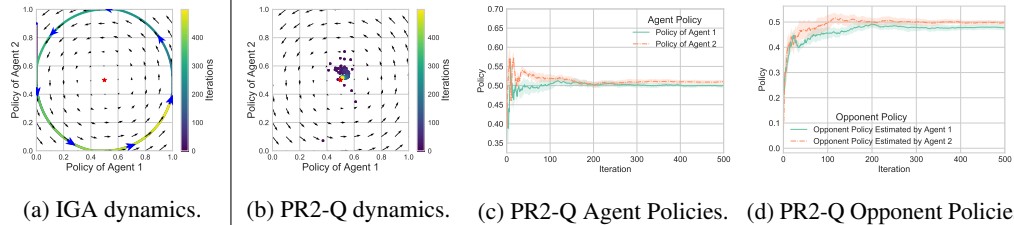

(a) IGA dynamics.    (b) PR2-Q dynamics.    (c) PR2-Q Agent Policies.    (d) PR2-Q Opponent Policies

Figure 3: Learning paths on the iterated matrix game. a: IGA. b-d: PR2-Q.

Theorem 1 states that the learning of $\rho_{\phi^{-i}}^{-i}(a^{-i}|s,a^i)$ can be further converted to minimizing the *KL*-divergence between the estimated policy $\rho_{\phi^{-i}}^{-i}$ and the *advantage* function: $D_{\mathrm{KL}}\left(\rho_{\phi^{-i}}^{-i}(a^{-i}|s,a^i)\|\exp(Q^i(s,a^i,a^{-i})-Q^i(s,a^i))\right)$. We can obtain a solution to Eq. 12 by maintaining two $Q$-functions, and then iteratively update them. We prove the convergence under self-play when there is one equilibrium. This leads to a fixed-point iteration that resembles value iteration.

**Theorem 2.** *In a symmetric game with only one equilibrium, and the equilibrium meets one of the conditions: 1) the global optimum, i.e.* $\mathbb{E}_{\pi_*}\left[Q_t^i(s)\right] \geq \mathbb{E}_{\pi}\left[Q_t^i(s)\right]$; *2) a saddle point, i.e.* $\mathbb{E}_{\pi_*}\left[Q_t^i(s)\right] \geq \mathbb{E}_{\pi^i}\mathbb{E}_{\pi_*^{-i}}\left[Q_t^i(s)\right]$ *or* $\mathbb{E}_{\pi_*}\left[Q_t^i(s)\right] \geq \mathbb{E}_{\pi_*^i}\mathbb{E}_{\pi^{-i}}\left[Q_t^i(s)\right]$; *where $Q_*$ and $\pi_*$ are the equilibrium value function and policy, respectively. The PR2 soft value iteration operator defined by:*

$$\mathcal{T}Q^i(s,a^i,a^{-i}) \triangleq r^i(s,a^i,a^{-i}) + \gamma\mathbb{E}_{s',a^{i\prime}\sim p_s,\pi^i}\left[\log\int_{a^{-i\prime}}\exp(Q^i(s',a^{i\prime},a^{-i\prime}))\,\mathrm{d}a^{-i\prime}\right], \quad (13)$$

*is a contraction mapping.*

*Proof. See Appendix D.* ∎

## 4.4 SAMPLING IN CONTINUOUS ACTION SPACE

In continuous controls, getting the actions from the opponent policy $\rho_{\phi^{-i}}^{-i}(a^{-i}|s,a^i)$ is challenging. In this work, we follow Haarnoja et al. (2017) to adopt the amortized Stein Variational Gradient Descent (SVGD) (Liu & Wang, 2016; Wang & Liu, 2016) in sampling from the soft Q-function. Compared to MCMC, Amortized SVGD is a computationally-efficient way to estimate $\rho_{\phi^{-i}}^{-i}(a^{-i}|s,a^i)$. Thanks to SVGD, agent $i$ is able to reason about potential consequences of opponent bavhaviors $\int_{a^{-i}}\pi_{\theta^{-i}}^{-i}(a^{-i}|s,a^i)Q^i(s,a^i,a^{-i})\,\mathrm{d}a^{-i}$, and finally find the corresponding best response.

## 5 EXPERIMENTS

We evaluate the performance of PR2 methods on the iterated matrix games, differential games, and particle world environment. Those games can by design have a non-trivial equilibrium that requires certain levels of intelligent reasonings between agents. We compared our algorithm with a series of baselines. In the matrix game, we compare against IGA (Infinitesimal Gradient Ascent) (Singh et al., 2000). In the differential games, the baselines from multi-agent learning algorithms are MASQL (Multi-Agent Soft-Q) (Wei et al., 2018) and MADDPG (Lowe et al., 2017). We also including independent learning algorithms implemented through DDPG (Lillicrap et al., 2015). To compare against traditional method of opponent modeling, we include one baseline that is based on DDPG but with one additional opponent modeling unit that is trained in an online and supervised way to learn the most recent opponent policy, which is then fed into the critic. Similar approach has been implemented by Rabinowitz et al. (2018) in realizing machine theory of mind. Besides, we applied centralized Symplectic Gradient Adjustment (SGA) (Balduzzi et al., 2018) optimization for DDPG agents (DDPG-SGA), which has recently been found to help converge to a local equilibrium quickly.

For the experiment settings, all the policies and $Q$-functions are parameterized by the MLP with 2 hidden layers, each with 100 units ReLU activation. The sampling network $\xi$ for the $\rho_{\phi^{-i}}^{-i}$ in SGVD follows the standard normal distribution. In the iterated matrix game, we trained all the methods including the baselines for 500 iterations. In the differential game, we trained the agents for 350 iterations with 25 steps per iteration. For the actor-critic methods, we set the exploration noise to 0.1 in first 1000 steps, and the annealing parameters for PR2-AC and MASQL are set to 0.5 to balance between the exploration and acting as the best response.

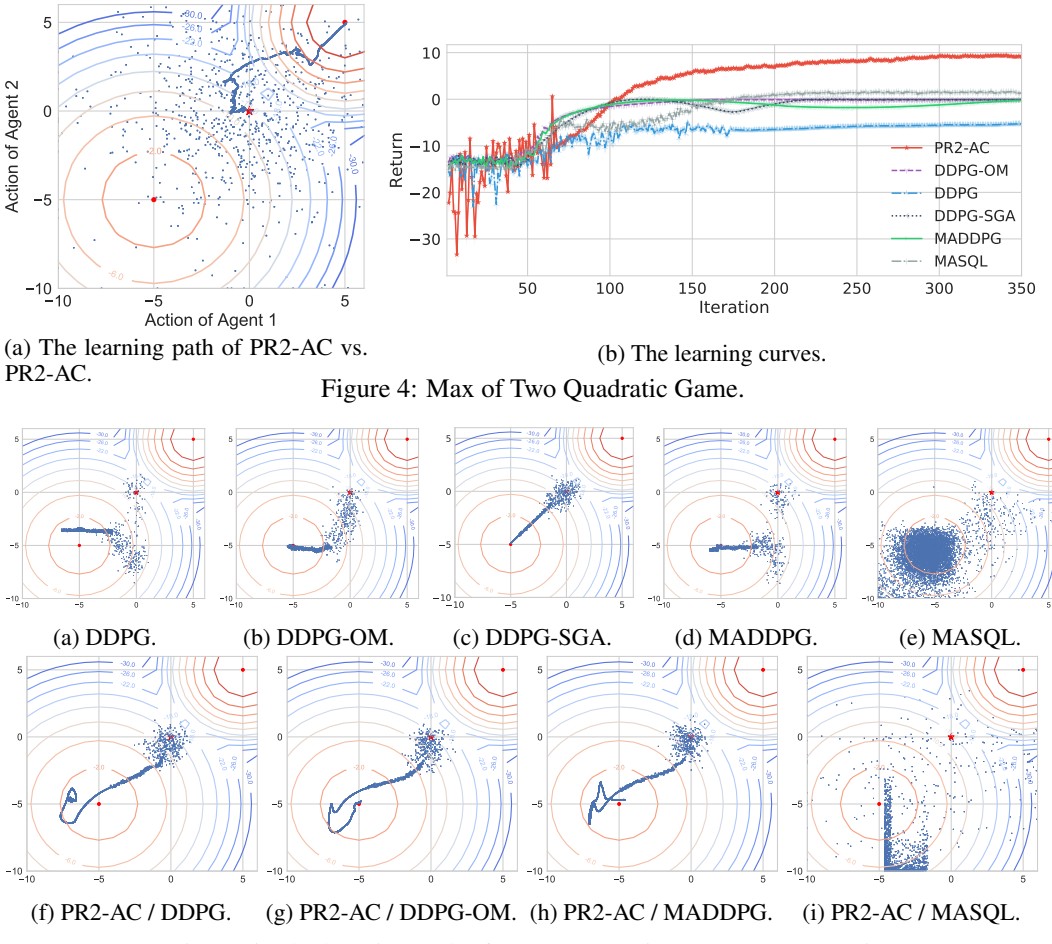

(a) The learning path of PR2-AC vs. PR2-AC.

(b) The learning curves.

Figure 4: Max of Two Quadratic Game.

(a) DDPG.  (b) DDPG-OM.  (c) DDPG-SGA.  (d) MADDPG.  (e) MASQL.

(f) PR2-AC / DDPG.  (g) PR2-AC / DDPG-OM.  (h) PR2-AC / MADDPG.  (i) PR2-AC / MASQL.

Figure 5: The learning path of Agent 1 (x-axis) vs. Agent 2 (y-axis).

## 5.1 ITERATED MATRIX GAME

In the matrix game, the payoffs are defined by: $R^1 = \begin{bmatrix} 0 & 3 \\ 1 & 2 \end{bmatrix}$, and $R^2 = \begin{bmatrix} 3 & 2 \\ 0 & 1 \end{bmatrix}$. These exists the only Nash Equilibrium at $(0.5, 0.5)$. This game has been intensively investigated in multi-agent studies (Bowling & Veloso, 2001a;b). One reason is that in solving the Nash Equilibrium for this game, simply taking simultaneous gradient steps on both agent's value functions will present the rotational behaviors on the gradient vector field; this leads to an endlessly iterative change of behaviors. Without considering the consequence of one agent's action on the other agent beforehand, it is challenging for both players to find the equilibrium. Similar issue has been found on training the GANs (Goodfellow et al., 2014; Mescheder et al., 2017)

The results are shown in Fig. 3. As expected, IGA fails to converge to the equilibrium but rotate around the equilibrium point. On the contrary, our method can find precisely the central equilibrium with a fully distributed fashion (see Fig. 3b). The convergence can also be justified by the agents' policies in Fig. 3c, and the opponent's policy that is maintained by each agent in Fig. 3d.

## 5.2 DIFFERENTIAL GAME

We adopt the same differential game, the Max of Two Quadratic Game, as Panait et al. (2006); Wei et al. (2018). The agents have continuous action space of $[-10, 10]$. Each agent's reward depends on the joint action following the equations: $r^1 (a^1, a^2) = r^2 (a^1, a^2) = \max (f_1, f_2)$, where $f_1 = 0.8 \times [-(\frac{a^1+5}{3})^2 - (\frac{a^2+5}{3})^2]$, $f_2 = 1.0 \times [-(\frac{a^1-5}{1})^2 - (\frac{a^2-5}{1})^2] + 10$. The task poses a great challenge to general gradient-based algorithms because gradient tends to points to the sub-optimal solution. The reward surface is shown in Fig. 4a; there is a local maximum 0 at $(-5, -5)$ and a global maximum 10 at $(5, 5)$, with a deep valley staying in the middle. If the agents' policies are initialized to $(0, 0)$ (the red starred point) that lies within the basin of the left local maximum, the gradient based methods would tend to fail to find the global maximum equilibrium point due to the

Figure 6: Performance of PR2-AC on the Particle World environment. Each bar shows the $0 - 1$ normalized score for agent in cooperative navigation task and the normalized advantage score (*agent reward - adversary reward*) in a set of competitive tasks. Higher score is better.

valley blocking the upper right area. The pathology of finding a suboptimal Nash equilibrium is also called *relative over-generalization* (Wei & Luke, 2016).

We present the results in Fig. 4b, PR2-AC shows superior performance that manages to converge to the global equilibrium, while all the other baselines fall into the local basin on the left, except that the MASQL has small chance to find the optimal point. On top of the convergence result, it is worth noting that as the temperature annealing is required for energy-based RL methods, the learning outcomes of MASQL are extremely sensitive to the way of annealing, i.e. when and how to anneal the temperature to a small value during training is non-trivial. However, our method does not need to tune the the annealing parameter at all because the each agent is acting the best response to the approximated conditional policy, considering all potential consequences of the opponent's response.

Interestingly, by comparing the learning path in Fig. 4a against Fig. 5(a-e) where the scattered blue dots are the exploration trails at the beginning, we can tell that if the PR2-AC model finds the peak point in joint action space, the agents can quickly go through the shortcut out of the local basin in a *clever* way, while other algorithms just converge to the local equilibrium. This further justifies the effectiveness and benefits of conducting recursive reasoning with opponents. Apart from testing in the self-play setting, we also test the scenario when the opponent type is different. We pair PR2-AC with all four baseline algorithms in Fig. 5(f-i). Similar result can be found, that is, algorithm that has the function of taking into account the opponents (i.e. DDPG_OM & MADDPG) can converge to the local equilibrium even though not global, while DDPG and MASQL completely fails due to the inborn defect from the independent learning methods.

## 5.3 PARTICLE WORLD ENVIRONMENTS

We further test our method on the multi-state multi-player Particle World Environments (Lowe et al., 2017). This includes four testing scenarios: 1) *Cooperative Navigation* with 3 agents and 3 landmarks. Agents are collectively rewarded based on the proximity of any agent to each landmark while avoiding collisions; 2) *Physical Deception* with 1 adversary, 2 good agents, and 2 landmarks. All agents observe the positions of landmarks and other agents. Only one landmark is the true target landmark. Good agents are rewarded based on how close any of them is to the target landmark, and how well they deceive the adversary; 3) *Keep-away* with 1 agent, 1 adversary, and 1 landmark. Agent is rewarded based on distance to landmark. Adversary is rewarded if it push away the agent from the landmark; 4) *Predator-prey* with 1 prey agent who moves faster try to run away from 3 adversary predator who move slower but are motivated to catch the prey cooperatively.

The PR2 methods are compared against a series of the centralized MARL methods in Fig. 6. Under the fully-cooperative setting (the left plot), PR2AC achieves the best performance over all baselines, even though it is a decentralized algorithm that does not have access to the exact opponent policies. Under the competitive settings where PR2AC rivals against the a set of adversary baselines, we find that PR2AC learners can beat all the baselines, including DDPG, DDPG-OM, and MASQL. The only exception is MADDPG, as it is suggested by the drop-down arrow. PR2AC performs particularly bad on the physical deception task. We believe it is mainly because the centralized critic can access the full knowledge of the exact policies of PR2-AC, but PR2-AC cannot access the models of its opponents in the reversed way; this could place PR2-AC in an inferior position during testing time as its deceptive strategy has been found out by the opponents already during training.

## 6 CONCLUSION

Inspired by the recursive reasoning capability of human intelligence, in this paper, we introduce a probabilistic recursive reasoning framework for multi-agent RL that follows "I believe that you believe that I believe". We adopt variational Bayes methods to approximate the opponents' conditional policy, to which each agent finds the best response and then improve their own policy. The training and execution is full decentralized and the resulting algorithms, PR2-Q and PR2-AC, converge in self-play when there is one Nash equilibrium. Our results on three kinds of testing beds with increasing complexity justify the advantages of learning to reason about the opponents in a recursive manner. In the future, we plan to investigate other approximation methods for the PR2 framework, and test our PR2 algorithm for the coordination task between AI agents such as coordinating autonomous cars before the traffic light.

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

APPENDIX

A    DECENTRALIZED MULTI-AGENT PROBABILISTIC RECURSIVE REASONING ALGORITHMS

Algorithm 1 gives the step by step learning procedures for PR2-AC algorithm.

---

**Algorithm 1:** Multi-Agent Probabilistic Recursive Reasoning Actor Critic (PR2-AC).

---

**Result:** Policy:$\pi^i$, Opponent Recursive Reasoning: $\rho^{-i}(a^{-i}|s, a^i)$.

1  Initialize parameters $\theta^i, \phi^{-i}, \omega^i$ for each agent $i$, and the random process $\mathcal{N}$ for action exploration.

2  Assign target parameters of joint action $Q$-function: $\omega^{i\prime} \leftarrow \omega^i$, and target policy parameter: $\theta^{i\prime} \leftarrow \theta^i$

3  $D^i \leftarrow$ empty replay buffer for each agent.

4  **for** *each episode* **do**

5  $\quad$ Initialize random process $\mathcal{N}$ for action exploration.

6  $\quad$ **for** *each step t* **do**

7  $\quad\quad$ Given the current $s$, for each agent $i$, select action $a^i = \mu^i_{\theta^i}(s) + \mathcal{N}_t$;

8  $\quad\quad$ Take the joint action $(a^i, a^{-i})$ and observe own reward $r^i$ and new state $s'$;

9  $\quad\quad$ Add the tuple $(s, a^i, a^{-i}, r^i, s')$ in corresponding replay buffer $D^i$;

10  $\quad\quad$ $s \leftarrow s'$;

11  $\quad\quad$ **for** *each agent i* **do**

12  $\quad\quad\quad$ Sample a random mini-batch $\{(s, a^i_j, a^{-i}_j, r^i_j, s'_j)\}^N_{j=0}$ from $D^i$;

13  $\quad\quad\quad$ Get $a^{i\prime}_j = \mu^{i\prime}_{\theta^i}$ for each state $s'_j$ ;

14  $\quad\quad\quad$ Sample $\{a^{-i}_{k,j}\}^M_{k=0} \sim \rho^{-i}_{\phi^{-i}}(\cdot|s'_j, a^{i\prime}_j)$ for each $a^{i\prime}_j$ and $s'_j$ ;

15  $\quad\quad\quad$ Set $y^i_j = r^i_j + \gamma \frac{1}{M} \sum^M_{k=0} Q^i_{\mu^{i\prime}}(s', a^{i\prime}, a^{-i\prime}_{k,j})$;

16  $\quad\quad\quad$ Update the critic by minimizing the loss $\mathcal{L}(\omega^i) = \frac{1}{N} \sum^N_{j=0} \left(y_j - Q^i_{\mu^i}(s_j, a^i_j, a^{-i}_j)\right)^2$;

17  $\quad\quad\quad$ Update the actor using the sampled policy gradient:

$$\nabla_{\theta^i} \eta^i \approx \frac{1}{N} \sum^N_{j=0} \nabla_{\theta^i} \mu^i(s_j) \nabla_{a^i} \frac{1}{M} \sum^M_{k=0} Q^i_{\mu^i}(s_j, a^i_j, a^{-i}_{k,j});$$

18  $\quad\quad\quad$ Compute $\Delta\rho^{-i}_{\phi^{-i}}$ using empirical estimation:

19

$$\Delta\rho^{-i}_{\phi^{-i}}(\cdot|s, a^i) = \mathbb{E}_{a^{-i}_t \sim \rho^{-i}_{\phi^{-i}}} \left[ \kappa\left(a^{-i}_t, \rho^{-i}_{\phi^{-i}}(\cdot; s_t, a^i_t)\right) \nabla_{\tilde{a}^{-i}} Q^i\left(s_t, a^i_t, \tilde{a}^{-i}\right)\big|_{\tilde{a}^{-i}=a^{-i}_t} \right.$$
$$\left. + \kappa\left(\tilde{a}^{-i}, \rho^{-i}_{\phi^{-i}}(\cdot; s_t, a^i_t)\right) \nabla_{\tilde{a}^{-i}}\big|_{\tilde{a}^{-i}=a^{-i}_t} \right],$$

20  $\quad\quad\quad$ where $\kappa$ is a kernel function;

21  $\quad\quad\quad$ Compute empirical gradient $\hat{\nabla}_{\phi^{-i}} J_{\rho^{-i}}$ ;

22  $\quad\quad\quad$ Update $\phi^{-i}$ according to $\hat{\nabla}_{\phi^{-i}} J_{\rho^{-i}}$;

23  $\quad\quad$ **end**

24  $\quad\quad$ Update target network parameters for each agent $i$:

25

$$\theta^{i\prime} \leftarrow \lambda\theta^i + (1-\lambda)\theta^{i\prime};$$
$$\omega^{i\prime} \leftarrow \lambda\omega^i + (1-\lambda)\omega^{i\prime};$$

26  $\quad$ **end**

27  **end**

---

The Algorithm 2 shows the variant of Decentralized Multi-Agent Probabilistic Recursive Reasoning. We can simply approximate the $\rho^{-i}(a^{-i}|s, a^i)$ by counting:$\rho^{-i}(a^{-i}|s, a^i) = C(a^i, a^{-i}, s)/C(a^i, s)$ in tabular if the state-action space is small, where $C$ is the counting function. It this case, an agent

---

**Algorithm 2:** Multi-Agent Probabilistic Recursive Reasoning $Q$-Learning (PR2-Q).

---

1 x **Result:** Policy: $\pi^i$, Opponent Recursive Reasoning: $\rho^{-i}(a^{-i}|s, a^i)$.
2 Initialize $Q^i(s, a^i, a^{-i})$ arbitrarily, set $\alpha$ as the learning rate, $\gamma$ as discount factor;
3 **while** *not converge* **do**
4     Given the current $s$, calculate the opponent best response $\rho^{-i}(a^{-i}|s, a^i)$ according to:
5

$$\rho^{-i}(a^{-i}|s, a^i) = \frac{1}{Z}\exp(Q^i(s, a^i, a^{-i}) - Q^i(s, a^i))$$

6     Select and sample action $a^i$ based on the Recursive Reasoning $\rho^{-i}(a^{-i}|s, a^i)$;
7

$$\text{softmax}(\int_{a^{-i}} \rho^{-i}(a^{-i}|s, a^i)Q^i(s, a^i, a^{-i}))$$

8     Observing joint-action $(a^i, a^{-i})$, reward $r^i$, and next state $s'$;
9

$$Q^i(s, a^i, a^{-i}) \leftarrow (1-\alpha)Q^i(s, a^i, a^{-i}) + \alpha(r^i + \gamma V^i(s'))$$
$$Q^i(s, a^i) \leftarrow (1-\alpha)Q^i(s, a^i) + \alpha(r^i + \gamma V^i(s'))$$

    where,

$$V^i(s) = \max_{a^i} \int_{a^{-i}} \rho^{-i}(a^{-i}|s, a^i)Q^i(s, a^i, a^{-i})$$

10 **end**

---

only needs to learn a joint action $Q$-function, and if the game is static, our method would degenerate to Conditional Joint Action Learning (CJAL) (Banerjee & Sen, 2007).

## B MULTI-AGENT POLICY GRADIENT

### B.1 MULTI-AGENT NON-CORRELATED POLICY GRADIENT

Since $\pi_\theta\left(a^i, a^{-i}|s\right) = \pi^i_{\theta^i}\left(a^i\right)\pi^{-i}_{\theta^{-i}}\left(a^{-i}|,a^i\right) = \pi^{-i}_{\theta^{-i}}\left(a^{-i}|s\right)\pi^i_{\theta^i}\left(a^i|s, a^{-i}\right)$, $\pi_\theta\left(a^i, a^{-i}|s\right)$ can be factorized as $\pi^i_{\theta^i}(a^i|s)\pi^{-i}_{\theta^{-i}}(a^{-i}|s)$ if $a^i$ and $a^{-i}$ are non-correlated. We follow the policy gradient formulation (Sutton et al., 2000; Wei et al., 2018) using Leibniz integral rule and Fubini's theorem which can give us Multi-Agent Non-correlated Policy Gradient:

$$\eta^i = \int_s \int_{a^i} \int_{a^{-i}} \pi(a^i, a^{-i}|s)Q^i(s, a^i, a^{-i}) \, da^{-i} \, da^i \, ds$$
$$= \int_s \int_{a^i} \int_{a^{-i}} \pi^i(a^i|s)\pi^{-i}(a^{-i}|s)Q^i(s, a^i, a^{-i}) \, da^{-i} \, da^i \, ds \qquad (14)$$
$$= \int_s \int_{a^i} \pi^i(a^i|s) \int_{a^{-i}} \pi^{-i}(a^{-i}|s)Q^i(s, a^i, a^{-i}) \, da^{-i} \, da^i \, ds.$$

Suppose the $\pi^i(a^i)$ is parameterized by $\theta^i$, and we apply the gradient over the $\eta^i$:

$$\nabla_{\theta^i}\eta^i = \int_s \int_{a^i} \nabla_{\theta^i}\pi^i_{\theta_i}(a^i|s) \int_{a^{-i}} \pi^{-i}(a^{-i}|s)Q^i(s, a^i, a^{-i}) \, da^{-i} \, da^i \, ds$$
$$= \mathbb{E}_{s\sim p, a^i\sim\pi^i}[\nabla_{\theta^i}\log\pi^i(a^i|s) \int_{a^{-i}} \pi^{-i}(a^{-i}|s)Q^i(s, a^i, a^{-i}) \, da^{-i}]. \qquad (15)$$

In practice, off-policy is more data-efficient. In MADDPG (Lowe et al., 2017) and COMA (Foerster et al., 2017), the replay buffer is introduced in a centralized deterministic actor-critic method for off-policy training. They apply batch sampling to the centralized critic which gives the joint-action $Q$-values:

$$\nabla_{\theta^i}\eta^i = \mathbb{E}_{s, a^i, a^{-i}\sim D}[\nabla_{\theta^i}\mu^i_{\theta^i}(a^i|s)\nabla_{a^i}Q^i(s, a^i, a^{-i})|_{a^i=\mu^i(s)}]. \qquad (16)$$

## B.2 MULTI-AGENT RECURSIVE REASONING POLICY GRADIENT

**Proposition 1.** *In a stochastic game, under the recursive reasoning framework defined by Eq. 3, the update rule for the multi-agent recursive reasoning policy gradient method can be devised as follows:*

$$\nabla_{\theta^i}\eta^i = \mathbb{E}_{s\sim p, a^i \sim \pi^i}\left[\nabla_{\theta^i}\log\pi_{\theta^i}^i(a^i|s)\int_{a^{-i}}\pi_{\theta^{-i}}^{-i}(a^{-i}|s,a^i)Q^i(s,a^i,a^{-i})\,\mathrm{d}a^{-i}\right]. \quad (17)$$

*Proof: As following.*

If we apply the chain rule to factorize the joint policy to: $\pi_\theta(a^i,a^{-i}|s) = \pi_{\theta^i}^i(a^i|s)\pi_{\theta^{-i}}^{-i}(a^{-i}|s,a^i)$. Then, we can have multi-agent recursive reasoning objective function as:

$$\begin{aligned}
\eta^i &= \int_s\int_{a^i}\int_{a^{-i}}\pi(a^i,a^{-i}|s)Q^i(a^i,a^{-i})\,\mathrm{d}a^{-i}\,\mathrm{d}a^i\,\mathrm{d}s \\
&= \int_s\int_{a^i}\pi^i(a^i|s)\int_{a^{-i}}\pi^{-i}(a^{-i}|s,a^i)Q^i(s,a^i,a^{-i})\,\mathrm{d}a^{-i}\,\mathrm{d}a^i\,\mathrm{d}s.
\end{aligned} \quad (18)$$

Compare to Eq. 14, $a^{-i}$ in Eq. 18 is additionally conditioned on $a^i$. We introduce agent $i$'a action $a^i$ into other agents's policies, leading to $\pi^{-i}(a^{-i}|s,a^i)$. We now compute the policy gradient analytically. Following the single agent Policy Gradient Theorem with Leibniz integral rule and Fubini's theorem, we get the multi-Agent Recursive Reasoning Policy Gradient:

$$\nabla_{\theta^i}\eta^i = \mathbb{E}_{s\sim p, a^i\sim\pi^i}[\nabla_{\theta^i}\log\pi^i(a^i|s)\int_{a^{-i}}\pi^{-i}(a^{-i}|s,a^i)Q^i(s,a^i,a^{-i})\,\mathrm{d}a^{-i}]. \quad (19)$$

However, in practice, the agent may not get access to other agents' policies. We need to infer the other agents' policies. We let $\rho_{\phi_{-i}}^{-i}(a^{-i}|s,a^i)$ denotes the parameterized opponent conditional policy of agent $i$ to approximate other agents policies, i.e, $\pi^{-i}(a^{-i}|s,a^i)$. Then we have Decentralized Multi-Agent Recursive Reasoning Policy Gradient comes as:

$$\begin{aligned}
\nabla_{\theta^i}\eta^i &\approx \mathbb{E}_{s\sim p,a^i\sim\pi^i}[\nabla_{\theta^i}\log\pi_{\theta^i}^i(a^i|s)\int_{a^{-i}}\rho_{\phi_{-i}}^{-i}(a^{-i}|s,a^i)Q^i(s,a^i,a^{-i})\,\mathrm{d}a^{-i}] \\
&= \mathbb{E}_{s\sim p,a^i\sim\pi^i}[\nabla_{\theta^i}\log\pi_{\theta^i}^i(a^i|s)Q_{\rho_{\phi_{-i}}^{-i}}^i(s,a^i)].
\end{aligned} \quad (20)$$

In Eq. 20, the gradient for agent $i$ is scaled by $Q_{\rho_{\phi_{-i}}^{-i}}^i(s,a^i) = \int_{a^{-i}}\rho_{\phi_{-i}}^{-i}(a^{-i}|s,a^i)Q^i(s,a^i,a^{-i})\,\mathrm{d}a^{-i}$. The trajectories generated by updated policy would help to train $\rho_{\phi_{-i}}^{-i}(a^{-i}|s,a^i)$ and $Q^i(s,a^i,a^{-i})$. These steps form a Expectation-Maximization style learning procedures: first, fix $\rho_{\phi_{-i}}^{-i}$ and $Q^i(s,a^i,a^{-i})$ to improve $\pi_{\theta^i}^i(a^i|s)$; then, improve $\rho_{\phi_{-i}}^{-i}$ and $Q^i(s,a^i,a^{-i})$ by the trajectories generated by $\pi_{\theta^i}^i(a^i|s)$. Furthermore, since PR2 method do not require opponents' actual private policies, Decentralized Multi-Agent Recursive Reasoning Policy Gradient can be decoupled from other agents' on-policies or target policies. In other words, the training can be conducted in an off-policy fashion by sampling mini-batches from the memory buffer $D$ with the help of the learned $\rho_{\phi_{-i}}^{-i}(a^{-i}|s,a^i)$ from $Q^i(s,a^i,a^{-i})$. ∎

## C OPPONENT CONDITIONAL POLICY INFERENCE VIA OPTIMAL TRAJECTORY

**Theorem 1.** *The optimal Q-function for agent $i$ that satisfies minimizing Eq. 10 is formulated as:*

$$Q_{\pi_\theta}^i(s,a^i) = \log\int_{a^{-i}}\exp(Q_{\pi_\theta}^i(s,a^i,a^{-i}))\,\mathrm{d}a^{-i}. \quad (21)$$

*And the corresponding optimal opponent conditional policy reads:*

$$\rho_{\phi^{-i}}^{-i}(a^{-i}|s,a^i) = \frac{1}{Z}\exp(Q_{\pi_\theta}^i(s,a^i,a^{-i}) - Q_{\pi_\theta}^i(s,a^i)) \quad (22)$$

*Proof. As following.*

Follow the proof in Levine (2018); Haarnoja et al. (2017), we first give the overall distribution by:

$$p(\tau) = [p(s_1) \prod_{t=1}^{T} p(s_{t+1}|s_t, a_t^i, a_t^{-i})] \exp(\sum_{t=1}^{T} r^i(s_t, a_t, a_t^{-i})). \tag{23}$$

We can adopt an optimization-based approach to approximate the opponent conditional policy, in which case the goal is to fit an approximation $\pi(a_t^i, a_t^{-i}|s_t) \approx \pi^i(a_t^i|s_t)\rho^{-i}(a_t^{-i}|s_t, a_t^i)$ such that the trajectory distribution,

$$\hat{p}(\tau) = p(s_1) \prod_{t=1}^{T} p(s_{t+1}|s_t, a_t^i, a_t^{-i}) \pi_{\theta^i}^i(a_t^i|s_t) \rho_{\theta^{-i}}^{-i}(a_t^{-i}|s_t, a_t^i), \tag{24}$$

has high likelihood to be observed. In the case of exact inference, as derived in the previous section, $D_{\mathrm{KL}}(\hat{p}(\tau)\|p(\tau)) = 0$. We can therefore view the inference process as minimizing the $KL$-divergence:

$$D_{\mathrm{KL}}(\hat{p}(\tau)\|p(\tau)) = -\mathbb{E}_{\tau \sim \hat{p}(\tau)}[\log p(\tau) - \log \hat{p}(\tau)]. \tag{25}$$

Negating both sides and substituting, we get:

$$
\begin{aligned}
-D_{\mathrm{KL}}(\hat{p}(\tau)\|p(\tau)) &= \mathbb{E}_{\tau \sim \hat{p}(\tau)}[\log p(s_1) + \sum_{t=1}^{T}(\log p(s_{t+1}|s_t, a_t, a_t^{-i}) + r^i(s_t, a_t^i, a_t^{-i})) \\
&\quad - \log p(s_1) - \sum_{t=1}^{T}(\log p(s_{t+1}|s_t, a_t^i, a_t^{-i}) + \log \pi(a_t^i, a_t^{-i}|s_t))] \\
&= \mathbb{E}_{\tau \sim \hat{p}(\tau)}[\sum_{t=1}^{T} r^i(s_t, a_t^i, a_t^{-i}) - \log \pi(a_t^i, a_t^{-i}|s_t)] \\
&= \sum_{t=1}^{T} \mathbb{E}_{(s_t, a_t^i, a_t^{-i}) \sim \hat{p}(s_t, a_t^i, a_t^{-i})}[r^i(s_t, a_t^i, a_t^{-i}) - \log \pi(a_t^i, a_t^{-i}|s_t)] \\
&= \sum_{t=1}^{T} \mathbb{E}_{(s_t, a_t^i, a_t^{-i}) \sim \hat{p}(s_t, a_t^i, a_t^{-i})}[r^i(s_t, a_t^i, a_t^{-i})] \\
&\quad + \mathbb{E}_{s_t, a_t^i \sim \hat{p}(s_t)}[\mathscr{H}(\rho^{-i}(a_t^{-i}|s_t, a_t^i))] + \mathbb{E}_{s_t \sim \hat{p}(s_t)}[\mathscr{H}(\pi^i(a_t^i|s_t))],
\end{aligned} \tag{26}
$$

where $\mathscr{H}$ is the entropy term. In the recursive case, we can rewrite the objective as follows:

$$Q^i(s, a^i) = \log \int_{a^{-i}} \exp(Q^i(s, a^i, a^{-i})) \, \mathrm{d}a^{-i}. \tag{27}$$

This corresponds to a standard bellman backup with a soft maximization for the value function. choosing optimal opponent recursive reasoning policy

$$\rho^{-i}(a^{-i}|s, a^i) = \frac{1}{Z} \exp(Q^i(s, a^i, a^{-i}) - Q^i(s, a^i)). \tag{28}$$

Then we can have the objective function:

$$
\begin{aligned}
J^i(\phi^{-i}) = \sum_{t=1}^{T} \mathbb{E}_{(s_t, a_t^i, a_t^{-i}) \sim \hat{p}(s_t, a_t^i, a_t^{-i})}[&r^i(s_t, a_t^i, a_t^{-i}) \\
&+ \mathscr{H}(\rho_{\phi^{-i}}^{-i}(a_t^{-i}|s_t, a_t^i)) + \mathscr{H}(\pi_{\theta^i}^i(a_t^i|s_t))].
\end{aligned} \tag{29}
$$

Then the gradient is then given by:

$$
\begin{aligned}
\nabla_{\phi^{-i}} J^i(\phi^{-i}) = \sum_{t=1}^{T} \mathbb{E}_{(s_t, a_t^i, a_t^{-i}) \sim p(s_t, a_t^i, a_t^{-i})}[\nabla_{\phi^{-i}} \log \rho_{\phi^{-i}}^{-i}(a_t^{-i}|s_t, a_t^i)(\sum_{t'=t}^{T} r^i(s_{t'}, a_{t'}^i, a_{t'}^{-i}))] \\
+ \nabla_{\phi^{-i}} \sum_{t=1}^{T} \mathbb{E}_{(s_t, a_t^i, a_t^{-i}) \sim p(s_t, a_t^i, a_t^{-i})}[\mathscr{H}(\rho_{\phi^{-i}}^{-i}(a_t^{-i}|s_t, a_t^i)) + \mathscr{H}(\pi_{\theta^i}^i(a_t^i|s_t))].
\end{aligned} \tag{30}
$$

The gradient of the entropy terms is given by:

$$\nabla_{\phi^{-i}} \mathscr{H}(\rho_{\phi^{-i}}^{-i}) = -\nabla_{\phi} \mathbb{E}_{(s_t, a_t^i) \sim p(s_t, a_t^i, a_t^{-i})} [\mathbb{E}_{a_t^{-i} \sim \rho_{\phi^{-i}}^{-i}(a_t^{-i}|s_t, a_t^i)} [\log \rho_{\phi^{-i}}^{-i}(a_t^{-i}|s_t, a_t^i)]]$$

$$= -\mathbb{E}_{(s_t, a_t^i, a_t^{-i}) \sim p(s_t, a_t^i, a_t^{-i})} [\nabla_{\phi} \log \rho_{\phi^{-i}}^{-i}(a_t^{-i}|s_t, a_t^i)(1 + \log \rho_{\phi^{-i}}^{-i}(a_t^{-i}|s_t, a_t^i)].$$

(31)

We can do the same for $\nabla_{\phi^{-i}} \mathscr{H}(\pi_{\theta^i}^i)$, and substitute these back we have:

$$\nabla_{\phi^{-i}} J^i(\phi^{-i}) = \sum_{t=1}^{T} \mathbb{E}_{(s_t, a_t^i, a_t^{-i}) \sim p(s_t, a_t^i, a_t^{-i})} [\nabla_{\phi^{-i}} \log \rho_{\phi^{-i}}^{-i}(a_t^{-i}|s_t, a_t^i)$$

(32)

$$(\sum_{t'=t}^{T} r^i(s_{t'}, a_{t'}^i, a_{t'}^{-i}) - \log \rho_{\phi^{-i}}^{-i}(a_{t'}^{-i}|s_t, a_t^i) - \log \pi_{\theta^i}^i(a_t^i|s_t) - 1)].$$

The $-1$ comes from the derivative of the entropy terms, and replacing $-1$ with a state and self-action dependent baseline $b(s_{t'}, a_{t'}^i)$ we can obtain the approximated gradient for $\phi$:

$$\nabla_{\phi^{-i}} J^i(\phi^{-i}) = \sum_{t=1}^{T} \mathbb{E}_{(s_t, a_t^i, a_t^{-i}) \sim p(s_t, a_t^i, a_t^{-i})} [\nabla_{\phi} \log \rho_{\phi^{-i}}^{-i}(a_t^{-i}|s_t, a_t^i)$$

$$(\sum_{t'=t}^{T} r^i(s_{t'}, a_{t'}^i, a_{t'}^{-i}) - \log \rho_{\phi^{-i}}^{-i}(a_{t'}^{-i}|s_{t'}, a_{t'}^i) - \log \pi_{\theta^i}^i(a_{t'}^i|s_{t'}) - \underbrace{1}_{\text{baseline ignore}})]$$

$$\approx \sum_{t=1}^{T} \mathbb{E}_{(s_t, a_t^i, a_t^{-i}) \sim p(s_t, a_t^i, a_t^{-i})} [\nabla_{\phi^{-i}} \log \rho_{\phi^{-i}}^{-i}(a_t^{-i}|s_t, a_t^i)$$

$$(r^i(s_t, a_t^i, a_t^{-i}) - \underbrace{\log \pi_{\theta^i}^i(a_t^i|s_t)}_{Q_t^i(s_t,a_t^i)-V_t^i(s_t)} - \underbrace{\log \rho_{\phi^{-i}}^{-i}(a_t^{-i}|s_t, a_t^i)}_{Q_t^i(s_t,a_t^i,a_t^{-i})-Q_t^i(s_t,a_t^i)}$$

$$+ \underbrace{\sum_{t'=t+1}^{T} r^i(s_{t'}, a_{t'}^i, a_{t'}^{-i}) - \log \rho_{\phi^{-i}}^{-i}(a_{t'}^{-i}|s_{t'}, a_{t'}^i) - \log \pi_{\theta^i}^i(a_{t'}^i|s_{t'}))]}_{\approx Q_t^i(s_{t+1},a_{t+1}^i,a_{t+1}^{-i})}$$

(33)

$$= \sum_{t=1}^{T} \mathbb{E}_{(s_t, a_t^i, a_t^{-i}) \sim p(s_t, a_t^i, a_t^{-i})} [\nabla_{\phi} \log \rho_{\phi^{-i}}^{-i}(a_t^{-i}|s_t, a_t^i)$$

$$(r^i(s_{t'}, a_{t'}^i, a_{t'}^{-i}) + Q_t^i(s_{t+1}, a_{t+1}^i, a_{t+1}^{-i}) - Q_t^i(s_t, a_t^i, a_t^{-i}) + \underbrace{V_t^i(s_t)}_{\text{ignore}})]$$

$$= \sum_{t=1}^{T} \mathbb{E}_{(s_t, a_t^i, a_t^{-i}) \sim p(s_t, a_t^i, a_t^{-i})} [(\nabla_{\phi^{-i}} Q_t^i(s_t, a_t^i, a_t^{-i}) - \nabla_{\phi^{-i}} Q_t^i(s_t, a_t^i))$$

$$(r^i(s_{t'}, a_{t'}^i, a_{t'}^{-i}) + Q_t^i(s_{t+1}, a_{t+1}^i, a_{t+1}^{-i}) - Q_t^i(s_t, a_t^i, a_t^{-i}) + \underbrace{V_t^i(s_t)}_{\text{ignore}})]$$

$$= \sum_{t=1}^{T} \mathbb{E}_{(s_t, a_t^i, a_t^{-i}) \sim p(s_t, a_t^i, a_t^{-i})} [(\nabla_{\phi^{-i}} Q_t^i(s_t, a_t^i, a_t^{-i}) - \nabla_{\phi^{-i}} Q_t^i(s_t, a_t^i))$$

$$(\hat{Q}_t^i(s_t, a_t^i, a_t^{-i}) - Q_t^i(s_t, a_t^i, a_t^{-i}))],$$

where $\hat{Q}_t^i(s_t, a_t^i, a_t^{-i})$ is is an empirical estimate of the $Q$-value of the policy. ∎

## D  SOFT BELLMAN EQUATION AND SOFT VALUE ITERATION

**Theorem 2.** *In a symmetric game with only one equilibrium, and the equilibrium meets one of the conditions: 1) the global optimum, i.e.* $\mathbb{E}_{\pi_*}[Q_t^i(s)] \geq \mathbb{E}_{\pi}[Q_t^i(s)]$*; 2) a saddle point, i.e.*

$\mathbb{E}_{\pi_*}\left[Q_t^i(s)\right] \geq \mathbb{E}_{\pi^i}\mathbb{E}_{\pi_*^{-i}}\left[Q_t^i(s)\right]$ *or* $\mathbb{E}_{\pi_*}\left[Q_t^i(s)\right] \geq \mathbb{E}_{\pi_*^i}\mathbb{E}_{\pi^{-i}}\left[Q_t^i(s)\right]$; *where $Q_*$ and $\pi_*$ are the equilibrium value function and policy, respectively. The PR2 soft value iteration operator defined by:*

$$\mathcal{T}Q^i(s,a^i,a^{-i}) \triangleq r^i(s,a^i,a^{-i}) + \gamma\mathbb{E}_{s',a^{i\prime}\sim p_s,\pi^i}\left[\log\int_{a^{-i\prime}}\exp(Q^i(s',a^{i\prime},a^{-i\prime}))\,\mathrm{d}a^{-i\prime}\right], \quad (34)$$

*is a contraction mapping.*

*Proof. As following:*

Based on Eq. 11 & 12 in Theorem 1, we can have the PR2 soft value iteration rules shown as:

$$
\begin{aligned}
Q_\pi^i(s,a^i,a^{-i}) &= r^i(s,a^i,a^{-i}) + \gamma\mathbb{E}_{s'\sim p_s}\left[\mathcal{H}(\pi^i(a^i|s)\pi^{-i}(a^{-i}|s,a^i)) + \mathbb{E}_{a^{-i\prime}\sim\pi^{-i}(\cdot|s',a^{i\prime})}[Q_\pi^i(s',a^{i\prime},a^{-i\prime})]\right] \\
&= r^i(s,a^i,a^{-i}) + \gamma\mathbb{E}_{s'\sim p_s}\left[Q_\pi^i(s',a^{i\prime})\right].
\end{aligned}
\tag{35}
$$

Correspondingly, we define the soft value iteration operator $\mathcal{T}$:

$$\mathcal{T}Q^i(s,a^i,a^{-i}) \triangleq r^i(s,a^i,a^{-i}) + \gamma\mathbb{E}_{s',a^{i\prime}\sim p_s,\pi^i}\left[\log\int_{a^{-i\prime}}\exp(Q^i(s',a^{i\prime},a^{-i\prime}))\,\mathrm{d}a^{-i\prime}\right]. \quad (36)$$

In a symmetric game with either one global equilibrium or saddle equilibrium, it has been shown by Yang et al. (2018) (see condition 1&2 in Theorem 1) that the payoff at the equilibrium point is unique. This validates applying the similar idea in proving the contraction mapping of soft-value iteration operator in the single agent case (see Lemma 1 in Fox et al. (2016)). We include it here to stay self-contained.

We first define a norm on $Q$-values as $\|Q_1^i - Q_2^i\| \triangleq \max_{s,a^i,a^{-i}}|Q_1^i(s,a^i,a^{-i}) - Q_2^i(s,a^i,a^{-i})|$. Suppose $\varepsilon = \|Q_1^i - Q_2^i\|$, then

$$
\begin{aligned}
\log\int_{a^{-i\prime}}\exp(Q_1^i(s',a^{i\prime},a^{-i\prime}))\,\mathrm{d}a^{-i\prime} &\leq \log\int_{a^{-i\prime}}\exp(Q_2^i(s',a^{i\prime},a^{-i\prime}) + \varepsilon)\,\mathrm{d}a^{-i\prime} \\
&= \log\int_{a^{-i\prime}}\exp(\varepsilon)\exp(Q_2^i(s',a^{i\prime},a^{-i\prime}))\,\mathrm{d}a^{-i\prime} \qquad (37) \\
&= \varepsilon + \log\int_{a^{-i\prime}}\exp(Q_2^i(s',a^{i\prime},a^{-i\prime}))\,\mathrm{d}a^{-i\prime}
\end{aligned}
$$

Similarly, $\log\int_{a^{-i\prime}}\exp(Q_1^i(s',a^{i\prime},a^{-i\prime}))\,\mathrm{d}a^{-i\prime} \leq -\varepsilon + \log\int_{a^{-i\prime}}\exp(Q_2^i(s',a^{i\prime},a^{-i\prime}))\,\mathrm{d}a^{-i\prime}$. Therefore $\|\mathcal{T}Q_1^i - \mathcal{T}Q_2^i\| \leq \gamma\varepsilon = \gamma\|Q_1^i - Q_2^i\|$. $\blacksquare$

