# OpenReview forum: "Probabilistic Recursive Reasoning for Multi-Agent Reinforcement Learning"
_ICLR.cc/2019/Conference_

### Official Review · AnonReviewer2 · 2018-11-01
**Significant updates to the new version**

**Rating:** 7
**Confidence:** 4

**Review:**

The paper introduces a decentralized training method for multi-agent reinforcement learning, where the agents infer the policies of other agents and use the inferred models for decision making. The method is intuitively straightforward and the paper provides some justification for convergence. I think the underlying theory is okay (new but not too surprising, a lot of the connections can be made with single agent RL), but the paper would be much stronger with experiments that have more than two players, one state and one dimensional actions.

(
Update: the new version of the paper addresses most of my concerns. There are a lot more experiments, and I think the paper is good for ICLR.

However, I wonder if the reasoning for PR2 is limited to "self-play", otherwise Theorem 1 could break because of the individual Q_i functions will not be symmetric. This could limit the applications to other scenarios.

Also, maybe explain self-play mathematically to make the paper self contained?
)

1. From the abstract, " PR2-Q and PR2-Actor-Critic, that are proved to converge in the self-play scenario". Theorem 2 only shows that under relatively strong assumptions (e.g. single Nash equilibrium), the soft value iteration operator is a contraction. This seems to have little to do with the actual convergence of PR2-Q and PR2-AC, especially AC which uses gradient-based approach. Also here the "convergence" in the abstract seem to imply convergence to the (single) global optimal solution (as is shown in the experiments), for which I thought you cannot prove even for single agent AC -- the best you can do is to show that gradient norm converges to zero, which gives you a local optima. Maybe things are different with the presence of the (concave) entropy regularization?

2. Theorem 2 also assumes that the opponent model $\rho$ will find the global optimal solution (i.e. (11, 12) can be computed tractably). However, the paper does not discuss the case where $\rho$ or $Q_\theta$ in question is imperfect (similar to humans over/underestimate its opponents), which might cause the actual solution to deviate significantly from the (single) NE. This would definitely be a problem in more high-dimensional MARL scenarios. I wonder if one could extend the convergence arguments by extending Prop 2.

3. The experiments mostly demonstrates almost the simplest non-trivial Markov games, where it could be possible that (11, 12) are true for PR2. However, the effectiveness of the method have not been demonstrated in other (slightly higher-dimensional) environments, such as the particle environments in the MADDPG paper. It does not seem to be very hard to implement this, and I wonder if this is related to the approximation error in (11, 12). The success in such environments would make the arguments much stronger, and provide sound empirical guidance to MARL practitioners.

Minor points:
- Are the policies in question stationary? How is PR2 different from the case of single agent RL (conditioned on perfect knowledge of a stationary opponent policy)?
- I have a hard time understanding why PR2 would have different behavior than IGA even with full knowledge of the opponent policy, assuming each policy is updated with infinitesimally small (but same) learning rates. What is the shape of the PR2 optimization function wrt agent 1?
- I wonder if using 2 layer neural networks with 100 units each on a 1 dimensional problem is overkill.
- Figure 4(a): what are the blue dots?
- Does (11) depend on the amount of data collected from the opponents? If so, how?
- I would recommend combining Prop 1 and prop 2 to save space. Both results are straightforward to prove, but the importance sampling perspective might be useful.
- Have you tried to compare with SGA (Balduzzi et al) or Optimistic mirror descent?
- I am also curious about an ablation study over the components used to infer opponent policies. A much simpler case would be action-dependent baselines, which seem to implicitly use some information about the opponents.

---

> ### Author Response · Authors · 2018-11-14
> **additional experiments added with other detailed questions addressed (part 1/2)**
>
>
> (This is part 1/2)
>  === AnonReviewer2 ===
>
> We first thank the reviewer for the constructive reviews that would surely help improving the paper. The main issue is that the experiment appears to be insufficient. We have addressed the issue. Following the comments, we have included additional comparisons of the following baselines:
> SGA (Balduzzi et al, 2018),
> WoLF-IGA (Bowling and Veloso, 2002),
> LOLA (Foerster et al, 2018),
> on the iterated matrix game. We add one extra zero-sum matrix game: matching pennies. We further evaluate our PR2 methods on multi-state multiplayer tasks that are more complicated. They include four different cooperative and competitive scenarios, i.e.
> Cooperative navigation,
> Physical deception,
> Predator-prey,
> Keep-away.
> The above scenarios settings follow those in MADDPG (Lowe, Ryan, et al. 2017).
>
> Based on the Fig. 6 in the updated submission, our conclusion is that the proposed PR2 method can achieve the best performance over all baselines on the fully-cooperative tasks. On the competitive tasks, PR2 still performs much better than all the independent learners, but slightly worse than MADDPG. We believe the main reason is that MADDPG is centralized method that requires to know the opponent’s exact policies whereas PR2 are fully decentralized methods. Due to the nature of competition, centralized methods that can know opponent policies in advance will have great advantages on the performance than decentralized algorithms.
>
> Our code has anonymously been uploaded onto the github (https://github.com/ml3705454/mapr2).
>
>
> Question_1: “discuss the case where $\rho$ or $Q_\theta$ in question is imperfect”
>
> Answer_1: We fully agree with the reviewer that there is possibility that the approximation error coming from the opponent modeling $\rho$ could destroy the whole sampling trajectory, and drag both agents to move away from the equilibrium. This explains why our current convergence proof requires the (strong) assumption of single Nash equilibrium under self-play, and the proof is in fact only valid when the $\rho$ is very close to the global optimum, or the $\rho$ should find the exact opponent policy. However, we believe that this is a fundamental challenge that current MARL community faces, for example, Nash-Q learning (Hu, Junling, et al. 2003), MADDPG(Lowe, Ryan, et al. 2017), LOLA  (Foerster et al, 2018), or the most recent SGA (Balduzzi et al, 2018), they all assume each agent has access to the exact parameters of the opponent; however, in adversarial settings, the opponent’s parameters are typically obscured,  and have to be inferred from the opponent’s state-action trajectories, such as by behavior cloning. In this sense, the approximation error from the inferring policies of other agents is  inevitable. However, our work is still one step further upon those methods. While those methods lose the convergence guarantee when the opponent policy is not exact, we have proved in theory and demonstrated by experiments that, by adopting the variational inference on approximating the opponent conditional policy and its subsequent soft-Q updates, the PR2 methods has nice convergence property in the game with one Nash equilibrium, without knowing the opponent’s exact policy functions.
>
> In fact, we tried to understand the impact of the approximation error on the matrix game on Section 5.1, it seems that under the self-play scenario with one single equilibrium, the approximation error decreases as the central agent heads towards the equilibrium. We believe understanding the impact of approximation error of opponent policy becomes critical especially in the multi-state multi-player environment, and this leaves considerable space for future work.

---

> ### Author Response · Authors · 2018-11-14
> **additional experiments added with other detailed questions addressed (part 2/2)**
>
>
> (This is part 2/2)
>  === AnonReviewer2 ===
>
> Question_2: “demonstrated in other (slightly higher-dimensional) environments, such as the particle environments in the MADDPG paper.”
>
> Answer_2: We have managed to add all 4 particle games that are used in the MADDPG paper. Please find Fig. 6 in the updated paper. Our method can achieve the best performance in the fully-cooperative task, and comparative performance in the other three competitive tasks comparing to the MADDPG that is a centralized method. We also tested the adversarial setting in those three competitive tasks (physical deception, predator-prey, and keep-away), where our method can beat the independent methods (DDPG, DDPG-OM), and keep better performance. However, we did find that it is hard to outperform on the task of physical deception; we believe it is mainly due to the fact that during training, centralized critic can access the full knowledge of the exact policies of PR2-AC, but PR2-AC cannot access the models of its opponents in a reverse way; this could potentially place PR2-AC in an inferior position in the testing time as the way how PR2-AC would deceive the opponent is known by the opponent already.
>
>
> Question_3:”difference between PR2 methods with single-agent RL and IGA”
>
> Answer_3: Both the single-agent RL and IGA can be categorised into the case of non-correlated factorization on the joint policy, as it is described by Eq.2 in Section 3.1, whereas PR2 methods conduct a different way of decomposition on the joint policy, as shown by Eq 3 in Section 4.1. Even equipped with full knowledge of the opponent, PR2 is still markedly different from single-agent RL or IGA because PR2 takes into account opponent’s behavior in its own decision making, i.e. $\pi(a^i | s, a^-i)$, and assuming all the agents will do so, as such, a recursive loop is formed, while the former two methods do not have certain dependency in the policies. We believe having such dependency in agent’s policy is critical for multiagent reinforcement learning tasks; it is justified by three different types of experiments, including matrix games, different games, and cooperative-competitive games where IGA or independent learning simply fails.
>
>
> Question_4: “blue dots in Figure 4a”
>
> Answer_4: those are the 1000 exploratory steps in the start phase before training.
>
>
> Question_5: “Does (11) depend on the amount of data collected from the opponents? If so, how?”
>
> Answer_5: The results in the original submission was reported by sampling the opponent policy 16 times during each update. We have tried sampling 32 times during the rebuttal period and find no major difference on the performance.
>
>
> Question_6: “compare with SGA (Balduzzi et al) or Optimistic mirror descent”
>
> Question_6: We added the SGA experiment on the differential game in Section 5.2, where the centralised SGA optimization with the two independent DDPG agents. In Fig.4b and Fig.5c,  SGA can steadily converge to the local maximum as promised in original paper, but compared to PR2-AC, it still fails to find the global maximum in that differential game.
>
>
> Question_7: “ablation study on inferring the opponent policies”
>
> Question_7: Instead of setting rule-based opponent strategy as baselines, we believe it is more illustrative to conduct the ablation study in such a way that using PR2 method to play against MADDPG which requires to know the exact policy parameters and actions that PR2 will take. Such nature of MADDPG makes it a perfect baseline model to understand the opponent module of PR2 methods.  The effectiveness of the opponent modeling module in PR2 is evaluated by Fig. 5h on the differential game, and by Fig. 6 on the multi-state cooperative-competitive games.

---

### Official Review · AnonReviewer3 · 2018-11-03
**Interesting and sound ideas and algorithms, but experimental validation is weak**

**Rating:** 7
**Confidence:** 4

**Review:**


# Summary:
The paper proposes a new approach for fully decentralized training in multi-agent reinforcement learning, termed probabilistic recursive reasoning (PR2). The key idea is to build agent policies that take into account opponent best responses to each of the agent's potential actions, in a probabilistic sense. The authors show that such policies can be seen as recursive reasoning, prove convergence of the proposed method in self-play, a demonstrate it in a couple of iterated normal form games with non-trivial Nash equilibria where baselines fail to converge.

I believe the community will find intuitions, methods, and theory developed by the authors interesting. However, I find some parts of the argument somewhat questionable as well as experimental verification insufficient (see comments below).


# Comments and questions:

## Weaknesses in the experimental evaluation:
I find it hard to justify a fairly complex algorithm (even though inspired by cognitive science), when most of the simpler alternatives from the literature haven't been really tested on the same iterated games (the baselines in the paper are all simple gradient-based policy search methods).

In the introduction (paragraph 2), the authors point out potential limitations of previous opponent modeling algorithms, but never compare with them in experiments. If the claim is that other methods "tend to work only under limited scenarios" while PR2 is more general, then it would be fair to ask for a comprehensive comparison of PR2 vs alternatives in at least 1 such scenario. I would be interested to see how the classical family of "Win or Learn Fast" (WoLF) algorithms (Bowling and Veloso, 2002) and the recent LOLA (Foerster et al, 2018) compare with PR2 on the iterated matrix game (section 5.1).

Also, out of curiosity, it would be interesting to see how PR2 works on simple iterated matrix games, eg iterated Prisoner's dilemma.

## Regarding the probabilistic formulation (section 4.3)
Eq. 8 borrows a probabilistic formulation of optimality in RL from Levine (2018). The expression given in Eq. 8 is proportional to the probability of a trajectory conditional on that each step is optimal wrt the agent's reward r^i, i.e., not for p(\tau) but for p(\tau | O=1).

If I understand it correctly, by optimizing the proposed KL objective, we fit both \pi^i and \rho^{-i} to the distribution of *optimal trajectories* with respect to r^i reward. That makes sense in a cooperative setting, but the problem arises when opponent's reward r^{-i} is different from r^i, in which case I don't understand how \rho^{-i} happens to approximate the actual policy of the opponent(s). Am I missing something here?

A minor point: shouldn't \pi^{-i} in eq. 9 be actually \rho^{-i}? (The derivations in appendix C suggest that.)

## Regarding alternative approaches (section 4.5)
The authors point out intractability of trying to directly approximate \pi^{-i}. The argument here is a little unclear. Wouldn't simple behavioral cloning work? Also, could we minimize KL(\pi^{-i} || \rho^{-i}) instead of KL(\rho^{-i} || \pi^{-i})?

# Minor
- I might be misreading it, but the last sentence of the abstract seems to suggest that this paper introduces opponent modeling to MARL, which contradicts the first sentence of paragraph 2 in the introduction.
- It is very hard to read plots in Figure 3. Would be nice to have them in a larger format.

Overall, I find the paper interesting, but it would definitely benefit from more thorough experimental evaluation.

---

> ### Author Response · Authors · 2018-11-14
> **Additional experiment added in the updated version of the paper (part 1/2)**
>
>
> (This is Part 1/2)
>  === AnonReviewer3 ===
>
> We first thank the reviewer for the constructive reviews that would surely help improving the paper. The main issue is that the experiment appears to be insufficient. We have addressed the issue. Following the comments, we have included additional comparisons of the following baselines:
> SGA (Balduzzi et al, 2018),
> WoLF-IGA (Bowling and Veloso, 2002),
> LOLA (Foerster et al, 2018),
> on the iterated matrix game. We add one extra zero-sum matrix game: matching pennies. We further evaluate our PR2 methods on multi-state multiplayer tasks that are more complicated. They include four different cooperative and competitive scenarios, i.e.
> Cooperative navigation,
> Physical deception,
> Predator-prey,
> Keep-away.
> The above scenarios settings follow those in MADDPG (Lowe, Ryan, et al. 2017).
>
> Based on the Fig. 6 in the updated submission, our conclusion is that the proposed PR2 method can achieve the best performance over all baselines on the fully-cooperative tasks. On the competitive tasks, PR2 still performs much better than all the independent learners, but slightly worse than MADDPG. We believe the main reason is that MADDPG is centralized method that requires to know the opponent’s exact policies whereas PR2 are fully decentralized methods. Due to the nature of competition, centralized methods that can know opponent policies in advance will have great advantages on the performance than decentralized algorithms.
>
> Our code has anonymously been uploaded onto the github (https://github.com/ml3705454/mapr2).
>
> Question_1: “A comprehensive comparison of PR2 vs alternatives in at least 1 such scenario, e.g. “Win or Learn Fast" (WoLF) algorithms (Bowling and Veloso, 2002) and the recent LOLA (Foerster et al, 2018) compare with PR2 on the iterated matrix game (section 5.1).”
>
> Answer_1: We have further added both of them (i.e. WolF-IGA and LOLA) on the matrix game in sec 5.1 in Appendix E. Note that that although both WolF-IGA and LOLA  (Fig.7 ) can converge to the Nash equilibrium, both of them require additional information, either the exact the equilibrium/payoff information, or the exact opponent’s value function and its gradient, whereas our PR2 methods do not need either of these information. Although this matrix game is simple; we believe it can help us understand the learning dynamics of the conditional opponent policy (see Fig. 3), especially under the condition that it can converge to non-trivial equilibrium with only the information of historical actions.
>
>
> Question_2: “by optimizing the proposed KL objective, we fit both \pi^i and \rho^{-i} to the distribution of *optimal trajectories* with respect to r^i reward. That makes sense in a cooperative setting, but the problem arises when opponent's reward r^{-i} is different from r^i, in which case I don't understand how \rho^{-i} happens to approximate the actual policy of the opponent(s). ”
>
> Answer_2: Despite the high level similarity between single-agent energy-based RL framework and our probabilistic recursive reasoning framework, the fundamental graphical model is different (see Fig. 8 in Appendix E). The “most probable trajectory” in the multi-agent case, represented together by the variables {O, O^-i}, does not necessarily stand for the trajectory where each agent just chooses the action that will give him the maximum reward (namely the “optimal trajectory” in single-agent case), but rather some kinds of equilibrium that no one would want to deviate from. In the example of the matrix game in Section 5.1, both agents reach the Nash equilibrium at (0.5, 0.5) in the end, that is because agent 1 knows choosing the action 1 which gives the maximum reward 3 (at the same time assuming agent 2 choose action 2) will not last because agent 2 will simply defect to choose action 1 to avoid the case of reward 0 for itself; therefore, the trajectory of (action 1, action 2) is not optimal to agent 1 anymore after considering the consequent influence on agent 2’s action. Another example is to think about the prisoner’s dilemma, (cooperate, cooperate) is not a probable trajectory, because it is always agent’s interest to defect, thereby the {O=1, O^-i =1} will only occur at the (defect, defect) instead. To sum up,  {O, O^-i} describes the likelihood of certain trajectory being observed, in the multi-agent scenario, the goal of equilibrium certainly allows the case where the opponent reward is different from r^i.
>
> Theoretically speaking, we have also proved in Theorem 2 that PR2 methods converge in the games with either fully-cooperative equilibrium or fully-competitive equilibrium. In addition, we have further added one extra zero-sum game, i.e. matching penny, in Fig. 9 of Appendix E. PR2 methods present convergent results as expected.

---

> > ### Public Comment · (anonymous) · 2018-11-22
> > **Prisoner's dilemma results + small issues around presentation of LOLA.**
> >
> > I agree with the reviewer 3: It would be interesting to see results for the iterated prisoner's dilemma . I would also suggest adding matching pennies.
> > PR2 looks like a very promising algorithm and seeing results on these standard iterated matrix games will help understand more about its functioning.
> >
> > Two small points regarding LOLA:
> > 1) LOLA was developed and tested with opponent modelling, so access to the opponent's policy is not required. Please see the results in the original paper.
> > 2) "..pre-defined opponent strategies (e.g. Tit-fot-Tat in iterated Prisoner’s Dilemma (Foerster et al., 2018))..": This is inaccurate. Tit-for-Tat is an emergent strategy that LOLA agents end up discovering, rather than pre-defined.

---

> > ### Comment · AnonReviewer3 · 2018-12-08
> > **Thanks for clarifications.**
> >
> > Thanks for the comments and more thorough comparison. This addresses most of my concerns.
> >
> > As noted by someone in the comment earlier, LOLA does seem to work with opponent modeling. So I would avoid claiming that as an advantage of PR2.

---

> ### Author Response · Authors · 2018-11-14
> **Additional experiment added in the updated version of the paper (part 2/2)**
>
>
> (This is part 2/2)
>  === AnonReviewer3 ===
>
> Question_3: “The argument is unclear on the intractability of directly approximate \pi^{-i}, could we minimize KL(\pi^{-i} || \rho^{-i})”
>
> Answer_3: Our work assume no access to opponent’s exact policy or value functions; therefore, any expectation over \pi^{-i} would require an additional layer of modelling the \pi^{-i} first, which we are concerned of introducing more approximation errors on top of the variational inference.

---

### Official Review · AnonReviewer4 · 2018-11-13
**A good first step towards endowing deep reinforcement learning agents with recursive reasoning capabilities**

**Rating:** 8
**Confidence:** 3

**Review:**

The high-level problem this paper tackles is that of endowing RL agents with recursive reasoning capabilities in a multi-agent setting, based on the hypothesis that recursive reasoning is beneficial for the agents to converge to non-trivial equilibria.

The authors propose the probabilistic recursive reasoning (PR2) framework for an n-agent stochastic game. The conceptual difference between PR2 and non-correlated factorizations of the joint policy is that, from the perspective agent i, PR2 augments the joint policy of all agents by conditioning the policies of agent i's opponents on the action that agent i took. The authors derive the policy gradient for PR2 and show that it is possible to learn these action-conditional opponent policies via variational inference in addition to learning the policy and critic for agent i.

The proposed method is evaluated on two experiments: one is an iterated matrix game and the other is a differential game ("Max of Two Quadratics"). The authors show in the iterated matrix game that baselines with non-correlated factorization rotate around the equilibrium point, whereas PR2 converges to it. They also show in the differential game that PR2 discovers the global optimum whereas baselines with non-correlated factorizations do not.

This paper is clear, well-motivated, and well-written. I enjoyed reading it. I appreciated the connection to probabilistic reinforcement learning as a means for formulating the problem of optimizing the variational distribution for the action-conditional opponent policy and for making such an optimization practical. I also appreciated the illustrative choice of experiments that show the benefit of recursive reasoning.

Currently, PR2 provides a proof-of-concept of recursive reasoning in a multi-agent system where the true equilibrium is already known in closed form; it remains to be seen to what extent PR2 is applicable to multi-agent scenarios where the equilibrium the system is optimizing is less clear (e.g. GANs for image generation). Overall, although the experiments are still small scale, I believe this paper should be accepted as a first step towards endowing deep reinforcement learning agents with recursive reasoning capabilities.

Below are several comments.

1. Discussion of limitations: As the authors noted in the Introduction and Related Work, multi-agent reinforcement problems that attempt to model opponents' beliefs often become both expensive and impractical as the number of opponents (N) and the recursion depth (k) grows because such complexity requires high precision in the approximate the optimal policy. The paper can be made stronger with experiments that illustrate to what extent PR2 practically scales to problems with N > 2 or K > 1 in terms of how practical it is to train.
2. Experiment request: To what extent do the approximation errors affect PR2's performance? It would be elucidating for the authors to include an experiment that illustrates where PR2 breaks down (for example, perhaps in higher-dimensional problems).
3. Minor clarification suggestion: In Figure 1: it would be clearer to replace "Angle" with "Perspective."
4. Minor clarification suggestion: It would be clearer to connect line 18 of Algorithm 1 to equation 29 on Appendix C.
5. Minor clarification suggestion: In section 4.5: "Despite the added complexity" --> "In addition to the added complexity."
6. Minor clarification: How are the importance weights in equation 7 reflected in Algorithm 1?
7. Minor clarification: In equation 8, what is the significance of integrating over time rather than summing?
8. Minor clarification: There seems to be a contradiction in section 5.2 on page 9. "the learning outcomes of PR2-AC and MASQL are extremely sensitive to the way of annealing...However, our method does not need to tune the the annealing parameter at all..." Does "our method" refer to PR2-AC here?

---

> ### Author Response · Authors · 2018-11-14
> **additional experiment results added with limitation of PR2 method discussed**
>
>
>
>  === AnonReviewer4 ===
>
> We first thank the reviewer for the constructive reviews that would surely help improving the paper. The main issue is that the experiment appears to be insufficient. We have addressed the issue. Following the comments, we have included additional comparisons of the following baselines:
> SGA (Balduzzi et al, 2018),
> WoLF-IGA (Bowling and Veloso, 2002),
> LOLA (Foerster et al, 2018),
> on the iterated matrix game. We add one extra zero-sum matrix game: matching pennies. We further evaluate our PR2 methods on multi-state multiplayer tasks that are more complicated. They include four different cooperative and competitive scenarios, i.e.
> Cooperative navigation,
> Physical deception,
> Predator-prey,
> Keep-away.
> The above scenarios settings follow those in MADDPG (Lowe, Ryan, et al. 2017).
>
> Based on the Fig. 6 in the updated submission, our conclusion is that the proposed PR2 method can achieve the best performance over all baselines on the fully-cooperative tasks. On the competitive tasks, PR2 still performs much better than all the independent learners, but slightly worse than MADDPG. We believe the main reason is that MADDPG is centralized method that requires to know the opponent’s exact policies whereas PR2 are fully decentralized methods. Due to the nature of competition, centralized methods that can know opponent policies in advance will have great advantages on the performance than decentralized algorithms.
>
> Our code has anonymously been uploaded onto the github (https://github.com/ml3705454/mapr2).
>
>
> Question_1: “Discussion of limitations: experiments that illustrate to what extent PR2 practically scales to problems with N > 2 or K > 1 in terms of how practical it is to train.”
>
> Question_2:  “Experiment request: To what extent do the approximation errors affect PR2's performance? It would be elucidating for the authors to include an experiment that illustrates where PR2 breaks down”
>
> Answers to 1 & 2: We have added experiments on more complicated multi-state multi-player game where we have 2-4 agents, and more complex action space (5 dimensions). Please find the result shown in Fig. 6 in the updated paper. Our method can achieve the best performance in the fully-cooperative task, and comparative performance in the other three competitive tasks comparing to the MADDPG that is a centralized method and require to know the exact opponent policy during training, while our methods do not. We also tested the adversarial setting in those three competitive tasks (physical deception, predator-prey, and keep-away), where our method can beat the independent methods (DDPG, DDPG-OM), and keep better performance. However, we did find that it is hard to outperform the MADDPG;  we believe it is mainly due to the fact that during training, centralized critic can access the full knowledge of the exact policies of PR2-AC, but PR2-AC cannot access the models of its opponents in a reverse way; this could potentially place PR2-AC in an inferior position in the testing time as the way how PR2-AC would deceive the opponent is known by the opponent already. On the other hand, we believe PR2 methods will face more challenges when the number of agents scales to hundreds. As each agent’s policy is dependent on the opponent’s behaviors $\pi (a^i | s, a^1, a^2, … , a^99)$, such dependency will become very hard even to sample from.
>
>
> Question_3:. How are the importance weights in equation 7 reflected in Algorithm 1?
>
> Answer_3: Since PR2 methods assume no knowledge about the opponents policies, the Algorithm 1 does not actually use the importance weights, but learns an approximated conditional policy \rho_{-i}.
>
> All the other comments have been addressed in the updated paper directly. We thank the reviewer for pointing out.

---

> > ### Comment · AnonReviewer4 · 2018-11-26
> > **Revisions address most of my concerns**
> >
> > The new version of the paper addresses most of my concerns. For the final version of the paper, I would suggest that the authors emphasize that they focus only level-1 recursion in the introduction to be precise about the scope of their contribution. It is a good first step towards level-k recursion for k > 1, but because the PR2 framework as presented captures only level-1 recursion, stating upfront that the paper is concerned with level-1 recursion would avoid confusing the reader.

---

> > > ### Author Response · Authors · 2018-11-26
> > > **Thanks for the comments.**
> > >
> > > Emphasizing K=1 is a critical point that we would surely incorporate in the final version, we thank the reviewer for highlighting this.

---

### Meta-Review · Area_Chair1 · 2018-12-14

**Confidence:** 4
**Recommendation:** Accept (Poster)

**Metareview:**

Pros:
- novel idea of endowing RL agents with recursive reasoning
- clear, well presented paper
- thorough rebuttal and revision with new results

Cons:
- small-scale experiments

The reviewers agree that the paper should be accepted.